# What Is So Special about Quantum Clicks?

**DOI:** 10.3390/e22060602

**Published:** 2020-05-28

**Authors:** Karl Svozil

**Affiliations:** Institute for Theoretical Physics, Vienna University of Technology, Wiedner Hauptstrasse 8-10/136, A-1040 Vienna, Austria; svozil@tuwien.ac.at

**Keywords:** correlation polytope, Kochen-Specker theorem, bell inequality, klyachko inequality, pitowsky principle of indeterminacy, 03.65.Ca, 02.50.-r, 02.10.-v, 03.65.Aa, 03.67.Ac, 03.65.Ud

## Abstract

This is an elaboration of the “extra” advantage of the performance of quantized physical systems over classical ones, both in terms of single outcomes as well as probabilistic predictions. From a formal point of view, it is based on entities related to (dual) vectors in (dual) Hilbert spaces, as compared to the Boolean algebra of subsets of a set and the additive measures they support.

## 1. Quantum Hocus Pocus

Time after time, scientists in other areas as well as theologians, philosophers, and artists, articulate an interest in understanding and comprehending what is so special about quantum physics. They demand to know the gist of quantization and its novel capacities. I have witnessed that individual physicists, in order to cope with such inquiries, often respond with at least two extreme strategies, both amounting to a ‘why bother?’ approach [1]:

(I) The first strategy is to play the “magic (joker) card” and respond by claiming, “quantum mechanics is magic”.

Alas, such types of hocus pocus [2] tend to leave the enquirer in an uneasy state. Because, first of all, it is frustrating to accept incomprehensibility in physics proper; quasi at the very heart and inner sanctum of contemporary natural science. Secondly, theologians have a very understandable natural suspicion with regards to any claims of sanctity and consecration, as this is supposed to be their own bread-and-butter domain. They resent evangelical physicists [3] joining their own realm, and “throwing parties” there.

Two subvariants of this “magic card” optional response are (i) either the acknowledgement [4,5] that (Chapter 6 in [6]) “nobody understands quantum mechanics”. So in order not to “get ’down the drain’, into a blind alley from which nobody has yet escaped” it is suggested that one should avoid asking, “how can it be like that?”–(ii) or claims that quantum mechanics needs no (further) interpretations [7] and explanations [8].

(II) The second strategy to cope with quantum mechanics is a formalistic and nominalistic one: to develop the mathematical formalism, mostly Hilbert spaces, very often not exceeding a good undergraduate text [9], and functional analysis. This is sometimes concealed by the rather hefty discussions of applications involving solutions of differential equations which at desperate times, may even employ asymptotic divergent (perturbation) series [10,11,12].

This latter strategy is frustrating to mathematicians and philosophers alike; as to the former, it seems that one tries to convince them that quantum mechanics is either trivial or plagued by inconsistencies and divergences. The latter group of philosophers would not take formulae they mostly hardly understand as a satisfactory answer: they suspect that syntax can never substitute semantics.

Another conceivable claim is a humble one inspired by Popper [13] and Lakatos [14]: all our scientific knowledge is preliminary and transitory; we know very little, and what we presume to know changes constantly as we move on to new ideas and concepts. There is no recognizable “ontological convergence” of concepts which we tend to approximate as we progress, no “truth” we seem to approach. Hence all our theories are embedded in, and part of, the history of thought.

With these caveats or provisions we might state that the current quantum phenomena indicate that we are living in a “vector world” rather than in a world spanned by subsets of sets (that is, power sets), and naive set-theoretical operations among them. In particular, quantum logic [15] suggests that the logico-algebraic relations and operations among (quantum) propositions need to be represented in terms of linear vector space entities (and their duals) endowed with a scalar product. In certain situations, this is radically different and departing from the “old” Boolean (sub)algebra ways.

In what follows we shall investigate those departures from classicality. They will manifest themselves in various ways and forms, and both in single outcomes as well as probabilistically. Some forms have no direct operational realization, and all of them must be based on idealistic constructions involving counterfactual observables. In extreme cases, the assumption of their (formal or physical) existence would result in a complete contradiction.

## 2. General Principles for Object/Observable Construction

One needs to be aware [16] that our cognition is not a priori given but could be imagined as a self-sustained “emergent” construction of the neuronal activities in our body (mostly located in the brain), which is subjected and exposed to “experiences” by our organs, and by self-reflection. Even if there would exist an “ontological anchor out there” (aka real entities) our perception would be bound to epistemological constraints, a fact known already since antiquity (514a–520a in [17]). In particular, physical observables need to be understood not as an objective fact about some physical reality independent of our perception, but subject to mental constructions involving conventions and presumptions. What are the intuitive conditions for some phenomenon to be called “observable”—and likewise, for a collection of “stuff” to be termed “object”? In addition, what exactly is it that we “observe” when measuring such “observables?” on such “objects”?

What has been discussed so far suggests that any Ansatz, or rather hint or suggestion, of what might qualify as an “object” or “observable” cannot merely be based on physical quantities alone but should also involve the entire human cognitive apparatus. Therefore, the notion of object originates in stuff which gets organized (by some cognitive agent) in spatial-temporal lumps encountered in the environment of an individual or a species. (To this end evolutionary psychology, and the evolution of cognition in general, is relevant).

Indeed, from an evolutionary point of view it can be expected that at least in some instances, a spread might develop between “good” and “bad” representations/predictions—that is, what might be considered an appropriate representation of the (potentially dangerous) environment, and a more pragmatic superstition about it which has a selective advantage. Because for survival pragmatism might, at least in the short term and for “ancient” setting encountered in a savanna, be more favorable than a careful evaluation of a situation: sometimes “to run for an escape” yield san advantage “to evaluate risks of danger” [18,19,20,21].

Of course, one could argue that in the long run, careful analysis and reflection on, say, the “reality/existence of observables/object” (Kahneman’s System 2) offers more selective advantages than a mere (impulsive) “blink [22]/guts feeling” (Kahneman’s System 1). This is effectively what happens as science progresses: the concepts and entities/objects/observables involved become increasingly adapted to the phenomena. Often those evolving concepts are more abstract and formalized than old ones. This does not necessarily mean that a “conceptual convergence” evolves insofar as “new, progressive theories/representations” are not necessarily extensions of “old, degenerative theories/representations” [14].

Therefore, we have to be suspicious of our own perception and its twisted capacity to comprehend the phenomena. We may even come to the conclusion that pragmatism (Kahneman’s *System 1*) is “more effective” (eg, in terms of Occam’s razor) to conceptualize an observable; and yet, in the long run, this conceptualization might turn out to be degenerative [14] and a failure. Often, overconfidence indicates a lack of competence [23].

Prima facie, objects originally qualified evolutionary either as being:negative; that is, dangerous, such as poisonous snakes, predators, atmospheric phenomena; orpositive; that which qualifies as prey/loot/prize with respect to nutrition or joy, such as eating/drinking/reproducing/breathing.

From that perspective, Kant was partially right, in as much as he suggested that such evolutionary ingrained environmental notions, structures and patterns appear to be hard-wired into our cognition, and thus suggest themselves as being “evident”: by evolutionary selection and substantiation they “appear natural”. However, Kant was deceived by not realizing that this sort of “evidence” and “naturalness” might actually be very deceptive: the cognitive concepts we inherited from, or share with, the plant/animal world serve well for survival and present many obvious immediate advantages – alas they are only functional with respect to, and serve as relative responses to, the environmental challenges encountered in the evolutionary past of those species “inheriting” them.

This is why we often have not the least inclination to seriously indulge in the idea that all stuff is an empty vacuum, pierced by extensionless elementary particles, as modern-day science suggests: we just did not need this idea to survive (so far). However, for example, we needed the concept of “snake” to survive in the desert. (I actually had this inspiration while contemplating a rock painting of a shaman taming a yellow snake at Spitzkoppe, Damaraland, Namibia.)

So it could be speculated that early object constructions were not formed by (sub)conscious cognitive processes. They rather developed as response patterns already at the earliest stages in the evolution of stuff capable of some (even very restricted, non-universal) forms of “computation” and endowed with cognitive capacities: nerves, brains, adaptive cycles which gain an advantage over nonadaptive behaviors by being capable of reaction against danger and opportunity. What we do when performing an object construction—for instance, creating narratives of what kind of observables are operational, or models of our brain et cetera—is just a more or less sophisticated extension thereof. In particular,
What qualifies a lump of stuff to be subjected to object/observable construction and become “an object” or “an observable” is its function with respect to us: otherwise—that is if it does not kill us or we cannot eat it et cetera—we might as well not perceive it as an individual entity separate from the rest of the stuff surrounding us.One might also speculate that every cub or human infant reenacts this structuralization of the environment–which was previously perceived ubiquitous, as a whole and non-separated (cf. also Piaget) from the cognitive agent–the whole issue of “external” versus “internal” comes into mind.as a consequence we as scientists have to be aware of these “hard-wired” conceptualizations or object constructions we and our species grew up with as “evident”, which have served our species well, but which eventually are too rigid and non-adaptive to be useful for the upcoming (deo volente) progressive research programs of Nature.

Particular models and instances of object/observable construction can be given in terms of (intertwining) Boolean subalgebras, resulting in partition logics [24] and orthomodular structures such as quantum logic discussed later.

## 3. Context and Greechie Orthogonality Hypergraphs

Henceforth a context will be any Boolean (sub-)algebra of compatible propositions that represent simultaneously measurable observables. The terms context, block, maximal observable, basis, clique, and classical mini-universe will be used synonymously.

In classical physics, there is only a single context, namely the entire set of observables. There exist models such as partition logics [25,26,27] – realizable by Wright’s generalized urn model [28] or automaton logic [29,30,31,32], – which are still quasi-classical but have more than one, possibly intertwined, contexts. Two contexts are intertwined if they share one or more common elements. In what follows we shall only consider contexts which if at all, intertwine at a single atomic proposition.

For such configurations Greechie has proposed a kind of orthogonality hypergraph [33,34,35] in which
entire contexts (Boolean subalgebras, blocks) are drawn as smooth lines, such as straight (unbroken) lines, circles or ellipses;the atomic propositions of the context are drawn as circles; andcontexts intertwining at a single atomic proposition are represented as non-smoothly connected lines, broken at that proposition.

In Hilbert space realizations, the straight lines or smooth curves depicting contexts represent orthogonal bases (or, equivalently, maximal observables, Boolean subalgebras or blocks), and points on these straight lines or smooth curves represent elements of these bases; that is, two points on the same straight line or smooth curve represent two orthogonal basis elements. From dimension three onwards, bases may intertwine [36] by possessing common elements.

## 4. General Principles for Probabilities of Objects/Observables

In this section, a very brief review of probability theory in arbitrary setups will be given. These axioms or requirements apply to all systems—classical as well as quantized and even more exotic ones—and therefore are uniform. In what follows we shall only consider finite configurations. Every maximal set of mutually exclusive and mutually co-measurable (in quantum mechanical terms: non-complimentary) observables will be called context (cf. Section 3 for a detailed discussion).

In what follows we shall assume the following axioms:A1:classical (sub)sets of finite (possibly extended) propositional/observable structures entail Kolmogorovian-type probabilities. In particular, they imply that within one and the same context, the corresponding probabilities are
K1(non-negativity): non-negative real numbers;K2(unity): of unit measure; that is, the probability of the occurrence of a complete set of propositions/observables is one;K3(additivity): the probabilities of mutually exclusive events $\left\{E_{1},\ldots ,E_{n}\right\}$ add up; that is, the probability of occurrence of all of them is the sum of the probabilities of occurrence of all of them; that is, $P\left(E_{1}\vee \ldots \vee E_{n}\right)=P\left(E_{1}\right)+\cdots +P\left(E_{n}\right)$.A2:(extended unity): Suppose there are two contexts $C_{1}=\left\{E_{1},\ldots E_{m}\right\}$ and $C_{2}=\left\{F_{1},\ldots F_{n}\right\}$. Then the sum of the conditional probabilities of all the elements of the second context, relative to any single element of the first context, adds up to one [37].

The latter Axiom A2 deals with situations that are characterized by empirical logics with more than one Boolean sublogics. These sublogics need not necessarily be “connected” or intertwined at one or more elements; they can be isolated. A2 intuitively states that if one selects one particular outcome of an observable, then the sum of the relative probabilities of all outcomes of another observable with respect to the previously chosen outcome of the first observable, must be one.

## 5. Classical Predictions: Truth Assignments and Probabilities

In what follows, we shall study observables whose classical and quantum predictions do not coincide and differ in various escalation levels—from “not very much” to “total”. Such predictions will come in two varieties: (i) stochastic/probabilistic, requiring a lot of individual observations; as well as (ii) individual outcome specific. To be able to make classical predictions we need to develop classical probability theory and logic. These classical predictions need then to be related so quantized systems with similar observables, and the respective differences in predictions need to be quantified.

Classical truth assignments will be formalized by two-valued measures whose image is either 0 or 1, associated with the logical values true and false, respectively. Classical probabilities and expectations will be introduced as the convex combination of “extreme” cases–associated with allowed (e.g., consistent)–which will be formalized by two-valued measures.

An upfront caveat: The alleged physical “existence” (ontology) of more than one context need not be—and, in general, due to quantum complementarity, for quanta—is not operational. That is, “most of the alleged “observables” in those collections of “observables” are not simultaneously measurable on any single individual particle (in any state). Therefore, as has already been pointed out by Specker [38], the following arguments make essential use of counterfactuals. Such counterfactuals are idealistic constructions of the mind that are identified with observables which could in principle have been measured yet have not been measured since the experimenter chose to measure another complementary observable. The earlier discussion in Section 2 of the construction of objects and physical observables is particularly pertinent to counterfactuals because one should not take for granted that all conceivable observables are defined simultaneously. Indeed, if such an assumption—the simultaneous existence of complementary/counterfactual objects of physical reality/observables is abandoned the entire chain of argument henceforth developed breaks down.

### 5.1. Truth Assignments as Two-Valued Measures, Frame Functions and Admissibility of Probabilities

In what follows we shall use notions of “truth assignments” on elements of logics which carry different names for related concepts:The (quantum) logic community uses the term *two-valued state;* or, alternatively, *valuation* for a *total* function *v* on all elements of some logic *L* mapping v:L→[0,1] such that (Definition 2.1.1, p. 20 in [39])
(a)v(I)=1,(b)if $\left\{a_{i},i\in N\right\}$ is a sequence of mutually orthogonal elements in *L*—in particular, this applies to atoms within the same context (block, Boolean subalgebra)—then the two-valued state is additive on those elements $a_{i}$; that is, additivity holds:
(1)$$v\left(\underset{i\in N}{⋁}\right)=\sum_{i\in N}v\left(a_{i}\right).$$Gleason has used the term frame function [36] (p. 886) of weight 1 for a separable Hilbert space H as a total, real-valued (not necessarily two-valued) function *f* defined on the (surface of the) unit sphere of H such that if $\left\{a_{i},i\in N\right\}$ represents an orthonormal basis of H, then additivity
(2)$$\sum_{i\in N}f\left(a_{i}\right)=1.$$
holds for all orthonormal bases (contexts, blocks) of the logic based on H.A dichotomic total function v:L→[0,1] will be called strongly admissible if
SAD1within every context $C=\{a_{i},i\in N\}$, a single atom $a_{j}$ is assigned the value one: $v(a_{j})=1$; andSAD2all other atoms in that context are assigned the value zero: $v(a_{i}\neq a_{j})=0$. Physically this amounts to only one elementary proposition being true; the rest of them are false. (One may think of an array of mutually exclusively firing detectors.)SAD3Non-contextuality, stated explicitly: The value of any observable, and, in particular, of an atom in which two contexts intertwine, does not depend on the context. It is context-independent.To cope with value indefiniteness (cf. Section 5.7.3), a weaker form of admissibility was proposed [40,41,42,43] which is no total function but rather is a *partial* function which may remain undefined (indefinite) on some elements of *L*: A dichotomic partial function v:L→[0,1] will be called *admissible* if the following two conditions hold for every context *C* of *L*:
WAD1if there exists a a∈C with v(a)=1, then v(b)=0 for all b∈C\{a};WAD2if there exists a a∈C with v(b)=0 for all b∈C\{a}, then v(a)=1;WAD3the value assignments of all other elements of the logic not covered by, if necessary, successive application of the admissibility rules, are undefined and thus the atom remains value indefinite.

Unless otherwise mentioned (such as for contextual value assignments or admissibility discussed in Section 5.7.3) the quantum logical (I), Gleason type (II), strong admissibility (III) notions of two-valued states will be used. Such two-valued states (probability measures) are interpretable as (pre-existing) truth assignments; they are sometimes also referred to as a Kochen-Specker value assignment [44].

### 5.2. Boole’s Conditions of Possible Experience

Already George Boole pointed out that
(i)the classical probabilities of certain events, as well as(ii)the classical probabilities of their (joint) occurrence, formalizable by products of the former “elementary” probabilities (i),
are subject to *linear* constraints [45,46,47,48,49,50,51,52,53,54,55,56,57,58,59,60,61,62,63,64,65,66]. A typical problem considered by Boole is this [46] (p. 229): *“Let $p_{1},p_{2},\ldots ,p_{n}$ represent the probabilities given in the data. As these will in general not be the probabilities of unconnected events, they will be subject to other conditions than that of being positive proper fractions, …. Those other conditions will, as will hereafter be shown, be capable of expression by equations or inequations reducible to the general form $a_{1}p_{1}+a_{2}p_{2}+\cdots +a_{n}p_{n}+a\geq 0$, $a_{1},a_{2},\ldots ,a_{n},a$ being numerical constants which differ for the different conditions in question. These … may be termed the conditions of possible experience”.*

Spool forward a century to Bell’s derivation [67] of some bounds on classical joint probabilities which could be perceived as special cases of Boole’s “conditions of possible experience”. Those classical probabilities characterize a setup of classical observables. However, these classical observables also have a *quantum double* [68]: such a corresponding quantized system has a very similar empirical logic [69]; that is, its structure of observables closely resembles the classical setup. However, one difference to its classical counterpart is its limited operational capacities; in particular, complementarity: Because of complementarity this needs to be done by measuring subsets, or *contexts of mutually compatible observables* (possibly by Einstein-Podolsky-Rosen type [70] counterfactual inference)—one at a time; e.g., one after another—on different distinct subensembles prepared in the same state. The predictions of the quantized system based on quantum probabilities can be tested, thereby falsifying the classical Boole-Bell type predictions based on classical (joint) probabilities. Please note that as observed by Sakurai [71] (pp. 241–243) and Pitowsky (Footnote 13 in [72]) the present form of the “Bell inequalities” is due to Wigner [73,74].

Froissart [75,76] suggested a geometric interpretation of Boole’s linear “conditions of possible experience” (without explicitly mentioning Boole): In referring to a later paper by Bell [77], Froissart proposed a general constructive method to produce all “maximal” (in the sense of tightest) constraints on classical probabilities and correlations for certain experimentally realizable quantum mechanical configurations. This method uses all conceivable types of classical correlated outcomes, represented as matrices (or higher-dimensional objects) which are the vertices [75] (p. 243) *“of a polyhedron which is their convex hull. Another way of describing this convex polyhedron is to view it as an intersection of half-spaces, each one corresponding to a face. The points of the polyhedron thus satisfy as many inequations as there are faces. Computation of the face equations is straightforward but tedious”.* In Froissart’s perspective certain “optimal” Bell-type inequalities can be interpreted as defining half-spaces (“below-above”, “inside-outside”) which represent the faces of a convex correlation polytope.

Later Pitowsky pointed out the connection to Boole; in particular that any Bell-type inequality can be interpreted as Boole’s condition of possible experience [72,78,79,80,81,82]. Pitowsky does not quote Froissart but mentions [78] (p. 1556) that he had been motivated by a (series of) paper(s) by Garg and Mermin [83] (who incidentally did not mention Froissart either) on Farkas’ Lemma.

A very similar question had also been pursued by Chochet theory [84], Vorob’ev [85] and Kellerer [86,87], who inspired Klyachko [88], as neither one of the previous authors are mentioned. (To be fair, in the reference section of an unpublished previous paper [89] Klyachko mentions Pitowsky two times; one reference not being cited in the main text).

### 5.3. The Convex Polytope Method

The essence of the convex polytope method is based on the observation that any classical probability distribution can be written as a convex sum of all of the conceivable “extreme” cases. These “extreme” cases can be interpreted as classical truth assignments; or, equivalently, as two-valued states. A two-valued state is a function on the propositional structure of elementary observables, assigning any proposition the values “0” and “1” if they are (for a particular “extreme” case) “false” or “true”, respectively. “Extreme” cases are subject to criteria defined later in Section 5.1. The first explicit use [26,27,90,91] (see Pykacz [92] for early use of two-valued states) of the polytope method for deriving bounds using two-valued states on logics with intertwined contexts seems to have been for the pentagon logic, discussed in Section 5.5.2) and cat’s cradle logic (also called “Käfer”, the German word for “bug”, by Specker), discussed in Section 5.5.3.

More explicitly, suppose that there be as many, say, *k*, “weights” $\lambda_{1},\ldots ,\lambda_{k}$ as there are two-valued states (or “extreme” cases, or truth assignments, if you prefer this denominations). Then convexity demands that all of these weights are positive and sum up to one; that is,
(3)$$\begin{array}{c} \lambda_{1},\ldots ,\lambda_{k}\geq 0,\;\mathrm{and}\;\\ \lambda_{1}+\ldots +\lambda_{k}=1. \end{array}$$

Suppose that for any particular *i*th two-valued state (or the *i*th “extreme” case, or the *i*th truth assignment, if you prefer this denomination), all the, say, *m*, “relevant” terms—relevance here merely means that we want them to contribute to the linear bounds denoted by Boole as conditions of possible experience, as discussed in Section 5.3.2—are “lumped” or combined together and identified as vector components of a vector $|x_{i}\rangle$ in an *m*-dimensional vector space $R^{m}$; that is,
(4)$$\begin{array}{c} |x_{i}\rangle ={\left(\begin{array}{c} x_{i_{1}},x_{i_{2}},\ldots .x_{i_{m}} \end{array}\right)}^{⊺}. \end{array}$$

Note that any particular convex see Equation (Equation 3) combination
(5)$$\begin{array}{c} |w\left(\lambda_{1},\ldots ,\lambda_{k}\right)\rangle =\lambda_{1}|x_{1}\rangle +\cdots +\lambda_{k}|x_{k}\rangle \end{array}$$
of the *k* weights $\lambda_{1},\ldots ,\lambda_{k}$ yields a valid—that is consistent, subject to the criteria defined in Section 5.1—classical probability distribution, characterized by the vector $|w(\lambda_{1},\ldots ,\lambda_{k})\rangle$. These *k* vectors $|x_{1}\rangle ,\ldots ,|x_{k}\rangle$ can be identified with vertices or extreme points (which cannot be represented as convex combinations of other vertices or extreme points), associated with the *k* two-valued states (or “extreme” cases, or truth assignments). Let $V=\left\{|x_{1}\rangle ,\ldots ,|x_{k}\rangle \right\}$ be the set of all such vertices.

For any such subset *V* (of vertices or extreme points) of $R^{m}$, the convex hull is defined as the smallest convex set in $R^{m}$ containing *V* (Section 2.10, p. 6 in [93]). Based on its vertices a convex V-polytope can be defined as the subset of $R^{m}$ which is the convex hull of a finite set of vertices or extreme points $V=\left\{|x_{1}\rangle ,\ldots ,|x_{k}\rangle \right\}$ in $R^{m}$:(6)$$\begin{array}{c} P=\mathrm{Conv}(V)=\\ =\left\{\sum_{i=1}^{k}\lambda_{i}|x_{i}\rangle |\lambda_{1},\ldots ,\lambda_{k}\geq 0,\;\sum_{i=1}^{k}\lambda_{i}=1,\;|x_{i}\rangle \in V\right\}. \end{array}$$

A convex H-polytope can also be defined as the intersection of a finite set of half-spaces, that is, the solution set of a finite system of *n* linear inequalities:(7)$$\begin{array}{c} P=P\left(A,b\right)=\left\{|x\rangle \in R^{m}|A_{i}|x\rangle \leq |b\rangle \;\mathrm{for}\;1\leq i\leq n\right\}, \end{array}$$
with the condition that the set of solutions is bounded, such that there is a constant *c* such that ∥|x〉∥≤c holds for all |x〉∈P. $A_{i}$ are matrices and |b〉 are vectors with real components, respectively. Due to the Minkoswki-Weyl “main” representation theorem [93,94,95,96,97,98,99] every V-polytope has a description by a finite set of inequalities. Conversely, every H-polytope is the convex hull of a finite set of points. Therefore the H-polytope representation in terms of inequalities as well as the V-polytope representation in terms of vertices, are equivalent, and the term convex polytope can be used for both and interchangeably. A *k*-dimensional convex polytope has a variety of faces which are again convex polytopes of various dimensions between 0 and k−1. In particular, the 0-dimensional faces are called vertices, the 1-dimensional faces are called *edges*, and the k−1-dimensional faces are called facets.

The solution of the hull problem, or the convex hull computation, is the determination of the convex hull for a given finite set of *k* extreme points $V=\{|x_{1}\rangle ,\ldots ,|x_{k}\rangle \}$ in $R^{m}$ (the general hull problem would also tolerate points inside the convex polytope); in particular, its representation as the intersection of half-spaces defining the facets of this polytope – serving as criteria of what lies “inside” and “outside” of the polytope—or, more precisely, as a set of solutions to a minimal system of linear inequalities. As long as the polytope has a non-empty interior and is full-dimensional (with respect to the vector space into which it is embedded) there are only inequalities; otherwise, if the polytope lies on a hyperplane one obtains also equations.

For the sake of a familiar example, consider the regular 3-cube, which is the convex hull of the 8 vertices in $R^{3}$ of $V=\{{\left(0,0,0\right)}^{⊺}$, ${\left(0,0,1\right)}^{⊺}$, ${\left(0,1,0\right)}^{⊺}$, ${\left(1,0,0\right)}^{⊺}$, ${\left(0,1,1\right)}^{⊺}$, ${\left(1,1,0\right)}^{⊺}$, ${\left(1,0,1\right)}^{⊺}$, ${\left(1,1,1\right)}^{⊺}\}$. The cube has 8 vertices, 12 edges, and 6 facets. The half-spaces defining the regular 3-cube can be written in terms of the 6 facet inequalities $0\leq x_{1},x_{2},x_{3}\leq 1$.

Finally, the correlation polytope can be defined as the convex hull of all the vertices or extreme points $|x_{1}\rangle ,\ldots ,|x_{k}\rangle$ in *V* representing the (*k* per two-valued state) “relevant” terms evaluated for all the two-valued states (or “extreme” cases, or truth assignments); that is,
(8)$$\begin{array}{c} \mathrm{Conv}\left(V\right)=\{|w\left(\lambda_{1},\ldots ,\lambda_{k}\right)\rangle |\\ ||w\left(\lambda_{1},\ldots ,\lambda_{k}\right)\rangle =\lambda_{1}|x_{1}\rangle +\cdots +\lambda_{k}|x_{k}\rangle \;,\\ \;\lambda_{1},\ldots ,\lambda_{k}\geq 0,\;\lambda_{1}+\ldots +\lambda_{k}=1,\;|x_{i}\rangle \in V\}. \end{array}$$

The convex H-polytope—associated with the convex V-polytope in (Equation 8)—which is the intersection of a finite number of half-spaces, can be identified with Boole’s conditions of possible experience.

A similar argument can be put forward for bounds on expectation values, as the expectations of dichotomic E∈{−1,+1}-observables can be considered to be affine transformations of two-valued states v∈{0,1}; that is, E=2v−1. One might even imagine such bounds on arbitrary values of observables, as long as affine transformations are applied. Joint expectations from products of probabilities transform non-linearly, as, for instance $E_{12}=\left(2v_{1}-1\right)\left(2v_{2}-1\right)=4v_{1}v_{2}-2\left(v_{1}+v_{2}\right)-1$. Therefore, given some bounds on (joint) expectations, these can be translated into bounds on (joint) probabilities by substituting $2v_{i}-1$ for expectations $E_{i}$. The converse is also true: bounds on (joint) probabilities can be translated into bounds on (joint) expectations by $v_{i}=\left(E_{i}+1\right)/2$.

This method of finding classical bounds must fail if, such as for Kochen-Specker configurations, there are no or “too few” (such that there exist two or more atoms which cannot be distinguished by any two-valued state) two-valued states. In this case, one may ease the assumptions; in particular, abandon admissibility, arriving at what has been called non-contextual inequalities [100].

#### 5.3.1. Why Consider Classical Correlation Polytopes when Dealing with Quantized Systems?

A caveat seems to be in order from the very beginning: in what follows correlation polytopes arise from classical (and quasi-classical) situations. The considerations are relevant for quantum mechanics only insofar as the quantum probabilities could violate classical bounds; that is if the quantum tests violate those bounds by “lying outside” of the classical correlation polytope.

There exist at least two good reasons to consider (correlation) polytopes for bounds on classical probabilities, correlations, and expectation values:they represent a systematic way of enumerating the probability distributions and deriving constraints—Boole’s conditions of possible experience—on them;one can be sure that these constraints and bounds are optimal in the sense that they are guaranteed to yield inequalities which are the best criteria for classicality.

It is not evident to see why, with the methods by which they have been obtained, Bell’s original inequality [77,101] or the Clauser-Horne-Shimony-Holt inequality [102] should be “optimal” at the time they were presented. Their derivation involves estimates which appear ad hoc, and it is not immediately obvious that bounds based on these estimates could not be improved. The correlation polytope method, on the other hand, offers a conceptually clear framework for a derivation of all classical bounds on higher-order distributions.

#### 5.3.2. What Terms May Enter Classical Correlation Polytopes?

What can enter as terms in such correlation polytopes? To quote Pitowsky [72] (p. 38), *“Consider n events $A_{1},A_{2},\ldots ,A_{n}$, in a classical event space … Denote $p_{i}=\mathrm{probability}\left(A_{i}\right)$, $p_{ij}=\mathrm{probability}\left(A_{i}\cap A_{j}\right)$, and more generally $p_{i_{1}i_{2}\cdots i_{k}}=\mathrm{probability}\left(A_{i_{1}}\cap A_{i_{2}}\cap \cdots \cap A_{i_{k}}\right)$, whenever $1\leq i_{1}<i_{2}<\ldots <i_{k}\leq n$. We assume no particular relations among the events. Thus $A_{1},\ldots ,A_{n}$ are not necessarily distinct, they can be dependent or independent, disjoint or non-disjoint etc”.*

However, although the events $A_{1},\ldots ,A_{n}$ may be in any relation to one another, one has to make sure that the respective probabilities, and, in particular, the extreme cases—the two-valued states interpretable as truth assignments—properly encode the logical or empirical relations among events. In particular, when it comes to an enumeration of cases, consistency must be retained. For example, suppose one considers the following three propositions: $A_{1}$: “it rains in Vienna”, $A_{3}$: “it rains in Vienna or it rains in Auckland”. It cannot be that $A_{2}$ is less likely than $A_{1}$; therefore, the two-valued states interpretable as truth assignments must obey $p\left(A_{2}\right)\geq p\left(A_{1}\right)$, and in particular, if $A_{1}$ is true, $A_{2}$ must be true as well. (It may happen though that $A_{1}$ is false while $A_{2}$ is true.) Also, mutually exclusive events cannot be true simultaneously.

These admissibility and consistency requirements are considerably softened in the case of non-contextual inequalities [100], where subclassicality–the requirement that among a complete (maximal) set of mutually exclusive observables only one is true and all others are false (equivalent to one important criterion for Gleason’s frame function [36])–is abandoned. To put it pointedly, in such scenarios, the simultaneous existence of inconsistent events such as $A_{1}$: “it rains in Vienna”, $A_{2}$: “it does not rain in Vienna” are allowed; that is, p(“itrainsinVienna”)=p(“itdoesnotraininVienna”)=1. The reason for this rather desperate step is that for Kochen-Specker type configurations, there are no classical truth assignments satisfying the classical admissibility rules; therefore the latter is abandoned. (With the admissibility rules goes the classical Kolmogorovian probability axioms even within classical Boolean subalgebras.)

It is no coincidence that most calculations are limited—or rather limit themselves because there are no formal reasons to go to higher orders–to the joint probabilities or expectations of just two observables: there is no easy “workaround” of quantum complementarity. The Einstein-Podolsky-Rosen setup [70] offers one for just two complementary contexts at the price of counterfactuals, but there seems to be no generalization to three or more complementary contexts in sight [103].

#### 5.3.3. General Framework for Computing Boole’s Conditions of Possible Experience

As pointed out earlier, Froissart and Pitowsky, among others such as Tsirelson, have sketched a very precise algorithmic framework for constructively finding all conditions of possible experience. In particular, Pitowsky’s later method [72,79,80,81,82], with slight modifications for very general non-distributive propositional structures such as the pentagon logic [26,27,91], goes like this:define the terms which should enter the bounds;(a)if the bounds should be on the probabilities: evaluate all two-valued measures interpretable as truth assignments;(b)if the bounds should be on the expectations: evaluate all value assignments of the observables;(c)if (as for non-contextual inequalities) the bounds should be on some pre-defined quantities: evaluate all such value definite pre-assigned quantities;arrange these terms into vectors whose components are all evaluated for a fixed two-valued state, one state at a time; one vector per two-valued state (truth assignment), or (for expectations) per value assignments of the observables, or (for non-contextual inequalities) per value-assignment;consider the set of all obtained vectors as vertices of a convex polytope;solve the convex hull problem by computing the convex hull, thereby finding the smallest convex polytope containing all these vertices. The solution can be represented as the half-spaces (characterizing the facets of the polytope) formalized by (in)equalities—(in)equalities which can be identified with Boole’s conditions of possible experience.

Froissart [75] and Tsirelson [76] are not very different; they arrange joint probabilities for two random variables into matrices instead of “delineating” them as vectors; but this difference is notational only. We shall explicitly apply the method to various configurations next.

The convex hull problem—finding the smallest convex polytope containig all these vertices, given a collection of such vertices—will be evaluated with Fukuda’s cddlib package cddlib-094h [104] (using GMP [105]) implementing the double description method [96,106,107]. The respective computer codes are listed in the Appendix A.

### 5.4. Non-Intertwined Contexts: Einstein-Podolsky-Rosen Type “Explosion” Setups of Joint Distributions

The first non-trivial (in the sense that the joint quantum probabilities and joint quantum expectations violate the classical bounds) instance occurs for four observables in an Einstein-Podolski-Rosen type “explosion” setup [70], where n>1 observables are measured on both sides, respectively.

Instead of a lengthy derivation of, say the Clauser-Horne-Shimony-Holt case of 2 observers, 2 measurement configurations per observer the reader is referred to standard computations thereof [72,79,81,91,108]. At this point, it might be instructive to realize how exactly the approach of Froissart and Tsirelson blends in [75,76]. The only difference to the Pitowsky method—which enumerates the (two-particle) correlations and expectations as vector components—is that Froissart and later and Tsirelson arrange the two-particle correlations and expectations as matrix components. For instance, Froissart explicitly mentions [75] (pp. 242, 243) 10 extremal configurations of the two-particle correlations, associated with 10 matrices
(9)$$\left(\begin{array}{cc} p_{13}=p_{1}p_{3} & p_{14}=p_{1}p_{4}\\ p_{23}=p_{2}p_{3} & p_{24}=p_{2}p_{4} \end{array}\right)$$
containing 0 s and 1 s (the indices “1, 2” and “3, 4” are associated with the two sides of the Einstein-Podolsky-Rosen “explosion”-type setup, respectively), arranged in Pitowsky’s case as vector
(10)$$\left(\begin{array}{c} p_{13}=p_{1}p_{3},p_{14}=p_{1}p_{4},p_{23}=p_{2}p_{3},p_{24}=p_{2}p_{4} \end{array}\right).$$

For probability correlations the number of different matrices or vectors is 10 (and not 16 as could be expected from the 16 two-valued measures), since, as enumerated in Table 1 some such measures yield identical results on the two-particle correlations; in particular, $v_{1},v_{2},v_{3},v_{4},v_{5},v_{9},v_{13}$ yield identical matrices (in the Froissart case) or vectors (in the Pitowsky case).

Going beyond the Clauser-Horne-Shimony-Holt case with 2 observers but more measurement configurations per observer is straightforward but increasingly demanding in terms of computational complexity [80]. The calculation for the facet inequalities for two observers and three measurement configurations per observer yields 684 inequalities [82,109,110]. If one considers (joint) expectations one arrives at novel ones which are not of the Clauser-Horne-Shimony-Holt type; for instance (p. 166, Equation (4) in [109]),
(11)$$\begin{array}{c} -4\leq -E_{2}+E_{3}-E_{4}-E_{5}+E_{14}-E_{15}+\\ +E_{24}+E_{25}+E_{26}-E_{34}-E_{35}+E_{36},\\ \mathrm{or}\\ -4\leq E_{1}+E_{2}+E_{4}+E_{5}+E_{14}+E_{15}+\\ +E_{16}+E_{24}+E_{25}-E_{26}+E_{34}-E_{35}. \end{array}$$

As already mentioned earlier, these bounds on classical expectations [109] translate into bounds on classical probabilities [82,110] (and vice versa) if the affine transformations $E_{i}=2v_{i}-1$ [and conversely $v_{i}=\left(E_{i}+1\right)/2$] are applied.

Here a word of warning is in order: if one only evaluates the vertices from the joint expectations (and not also the single particle expectations), one never arrives at the novel inequalities of the type listed in Equation (Equation 11), but obtains 90 facet inequalities; among them 72 instances of the Clauser-Horne-Shimony-Holt inequality form. They can be combined to yield (see also Ref. [109] p. 166, Equation (Equation 4)).
(12)$$\begin{array}{c} -4\leq E_{14}+E_{15}+E_{24}+E_{26}+E_{35}-E_{36}\leq 4. \end{array}$$

For the case of 3 [82] and more qubits, algebraic methods different than the hull problem for polytopes were suggested in Refs. [111,112,113,114].

### 5.5. Truth Assignments and Predictions for Intertwined Contexts

Let us first contemplate on the question or objection “why should we be bothered with the classical interpretation and probabilities of multiple contexts?” This could already have been asked for isolated contexts discussed earlier, and it becomes more compelling if intertwined contexts are considered. After all, “classical” empirical configurations require a single context—Boole’s algebra of observables”—only. Why consider the simultaneous existence of a multitude of those; and even more so if they are somehow connected and intertwined by assuming that one and the same observable may occur in different contexts?

A historic answer is this: “because with the advent of quantum complementarity we were confronted with such observables organized in distinct contexts, and we had to cope with them.” In particular, one could investigate the issue of whether or not such collections of quantum contexts and the observables therein would allow a classical interpretation relative to the assumptions made. Therefore, one crucial assumption was the *independence of the value of the observable from the particular context in which it appears*, a property often called non-contextuality. The adverse assumption [101,115] is often referred to as contextuality.

Another motivation for studying intertwined contexts comes from partition logic [26,27], and, in particular, from generalized urn models and finite automata. These cases still allow a quasi-classical interpretation although they are not strictly “classical” in the sense of a single Boolean algebra (although they allow a faithful, structure-preserving embedding into a single Boolean algebra).

In the following, we shall present a series of logics encountered by studying certain finite collections of quantum observables. The contexts (representable by maximal observables, Boolean subalgebras, blocks, or orthogonal bases) formed by those collections of quantum observables are intertwined; but “not very much” so: by assumption and for convenience, any two contexts intertwine in only one element; it does not happen that two contexts are pasted [33,34,39,116] along with two or more atoms. Such intertwines—connecting contexts by pasting them together—can only occur from Hilbert space dimension three onwards, because contexts in lower-dimensional spaces cannot have the same element unless they are identical.

Any such construction is usually based on a succession of auxiliary gadget graphs [117,118,119] stitched together to yield the desired properties. Therefore, gadgets are formed from gadgets of ever-increasing size and functional performance (see also Chapter 12 of Ref. [108]):0th order gadget: a single context (aka clique/block/Boolean (sub)algebra/maximal observable/orthonormal basis);1st order “firefly” gadget: two contexts connected in a single intertwining atom;2nd order gadget: two 1st order firefly gadgets connected in a single intertwining atom;3rd order house/pentagon/pentagram gadget: one firefly and one 2nd order gadget connected in two intertwining atoms to form a cyclic orthogonality diagram (hypergraph);4th order true-implies-false (TIFS)/01-(maybe better 10)-gadget: e.g., a Specker bug consisting of two pentagon gadgets connected by an entire context; as well as extensions thereof to arbitrary angles for terminal (“extreme”) points;5th order true-implies-true (TITS)/11-gadget: e.g., Kochen and Specker’s $\Gamma_{1}$, consisting of one 10-gadget and one firefly gadget, connected at the respective terminal points;6th order gadget: e.g., Kochen and Specker’s $\Gamma_{3}$, consisting of a combo of two 11-gadgets, connected by their common firefly gadgets;7th order construction: consisting of one 10- and one 11-gadget, with identical terminal points serving as constructions of Pitowsky’s principle of indeterminacy [43,120,121] and the Kochen-Specker theorem;

In Section 5.5.1 we shall first study the “firefly case” with just two contexts intertwined in one atom; then, in Section 5.5.2, proceed to the pentagon configuration with five contexts intertwined cyclically, then, in Section 5.5.3, paste two such pentagon logics to form a cat’s cradle (or, by another term, Specker’s bug) logic; and finally, in Section 5.6.1, connect two Specker bugs to arrive at a logic which has a so “meager” set of states that it can no longer separate two atoms. As pointed out already by Kochen and Specker (p. 70, Theorem 0 in [122]) this is no longer embeddable into some Boolean algebra. It thus cannot be represented by a partition logic, and thus has neither any generalized urn and finite automata models nor classical probabilities separating different events. The case of logics allowing no two-valued states will be covered consecutively.

#### 5.5.1. Probabilities on the Firefly Gadget

History: Cohen presented [123] (pp. 21–22) a classical realization of the first logic with just two contexts and one intertwining atom: a firefly in a box, observed from two sides of this box which are divided into two windows; assuming the possibility that sometimes the firefly does not shine at all. This firefly logic, which is sometimes also denoted by $L_{12}$ because it has 12 elements (in a Hasse diagram) and 5 atoms in two contexts defined by $\left\{a_{1},a_{2},a_{3}\right\}$ and $\left\{a_{3},a_{4},a_{5}\right\}$. In shorthand we may arrange the contexts, as well as the atomic propositios/observables containing them, in a set of sets; that is, $\left\{\left\{a_{1},a_{2},a_{3}\right\},\left\{a_{3},a_{4},a_{5}\right\}\right\}$. The Greechie orthogonality hypergraph of the firefly gadget is depicted in Figure 1a.

Classical interpretations: The five two-valued states on the firefly logic are enumerated in Table 2.

As long there are “sufficiently many” two-valued states—to be more precise, as long as there is a separating set of two-valued states (Theorem 0 in [122])–a canonical partition logic can be extracted from this set (modulo permutations) by an “inverse construction” [26,27]: because of admissibility constraints SAD1-SAD3 (cf. Section 5.1) every two-valued state has exactly one entry “1” per context, and all others “0”, by including the index *i* of the *i*’th measure $v_{i}$ in a subset of all indices of measures which acquire the value “1” on a particular atom one obtains a subset representation for that atom which is an element of the partition of the index set. This amounts to enumerating the set of all two-valued states as in Table 2 and searching its columns for entries “1”: in the firefly gadget case [25,27], this results in a canonical partition logic (modulo permutations) of
(13)$$\begin{array}{c} \{\{\{4,5\},\{2,3\},\{1\}\},\\ \{\{3,4\},\{2,5\},\{1\}\}\}. \end{array}$$

This canonical partition, in turn, induces all classical probability distributions, as enumerated in (Figure 12.4 in [108]).

Lovász [124,125] defined a faithful orthogonal representation [126] of a graph in some finite-dimensional Hilbert space by identifying vertices with vectors, and adjacency with orthogonality. No faithful orthogonal representation of the firefly gadget in $R^{3}$ is given here, but it is straightforward–just two orthogonal tripods with one identical leg will do (or can be read off from logics containing more such intertwined fireflies).

#### 5.5.2. Probabilities on the House/Pentagon/Pentagram

Admissibility of two-valued states imposes conditions and restrictions on the two-valued states already for a single context (Boolean subalgebra): if one atom is assigned the value 1, all other atoms have to have value assignment(s) 0. This is even more so for intertwining contexts. For the sake of an example, consider two firefly logics pasted along an entire block, in short $\{\left\{a_{1},a_{2},a_{3}\right\}$, $\left\{a_{3},a_{4},a_{5}\right\}$, $\left\{a_{5},a_{6},a_{7}\right\}\}$. The Greechie orthogonality hypergraph is depicted in Figure 1b (see also Figure 12.5 in [108]). For such logic we can state a “true-and-true implies true” rule: if the two-valued measure at the “outer extremities” is 1, then it must be 1 at its center atom.

History: We can pursue this path of ever-increasing restrictions through the construction of pasted; that is, intertwined, contexts. Let us proceed by stitching/pasting more firefly logics together cyclically in “closed circles”. The two simplest such pastings–two firefly logics forming either a triangle or a square Greechie orthogonal hypergraph–have no realization in three dimensional Hilbert space. The next diagram realizably is obtained by pasting three firefly logics. It is the house/pentagon/pentagram (the graphs of the pentagon and the pentagram are related by an isomorphic transformation of the vertices $a_{1}\mapsto a_{1}$, and $a_{9}\mapsto a_{5}\mapsto a_{3}\mapsto a_{7}\mapsto a_{9}$) logic also denoted as orthomodular house (p. 46, Figure 4.4 in [34]) and discussed in Ref. [60]; see also Birkhoff’s distributivity criterion (p. 90, Theorem 33 in [127]), stating that in particular, if some lattice contains a pentagon as a sublattice, then it is not distributive [128].

This cyclic logic is, in short, $\{\left\{a_{1},a_{2},a_{3}\right\}$, $\left\{a_{3},a_{4},a_{5}\right\}$, $\left\{a_{5},a_{6},a_{7}\right\},\left\{a_{5},a_{6},a_{7}\right\},\left\{a_{5},a_{6},a_{7}\right\}\}$. The Greechie orthogonality hypergraph of the house/pentagon/pentagram gadget is depicted in Figure 1c.

Classical interpretations: Such house/pentagon/pentagram logics allow “exotic” probability measures [129]: as pointed out by Wright [129] (p. 268) the pentagon allows 11 “ordinary” two-valued states $v_{1},\ldots ,v_{11}$, and one “exotic” dispersionless state $v_{e}$, which was shown by Wright to have neither a classical nor a quantum interpretation; all defined on the 10 atoms $a_{1},\ldots ,a_{10}$. They are enumerated in Table 3. These 11 “ordinary” two-valued states directly translate into the classical probabilities (Figure 12.4 in [108]).

The pentagon logic has quasi-classical realizations in terms of partition logics [25,26,27], such as generalized urn models [28,129] or automaton logics [29,30,31,32]. The canonical partition logic (up to permutations) is
(14)$$\begin{array}{c} \{\{\{1,2,3\},\{7,8,9,10,11\},\{4,5,6\}\},\\ \{\{4,5,6\},\{1,3,9,10,11\},\{2,7,8\}\},\\ \{\{2,7,8\},\{1,4,6,10,11\},\{3,5,9\}\},\\ \{\{3,5,9\},\{1,2,4,7,11\},\{6,8,10\}\},\\ \{\{6,8,10\},\{4,5,7,9,11\},\{1,2,3\}\}\}. \end{array}$$

The classical probabilities can be directly read off from the canonical partition logic: they are thee respective convex combination of the eleven two-valued states ($0\leq \lambda_{i}\leq 1$ and $\sum_{i=1}^{11}\lambda_{i}=1$): on $a_{1}$, $a_{2}$ and $a_{3}$ it is, for instance, $p_{1}=\lambda_{1}+\lambda_{2}+\lambda_{3}$, $p_{1}=\lambda_{7}+\lambda_{8}+\lambda_{9}+\lambda_{10}+\lambda_{11}$, $p_{3}=\lambda_{4}+\lambda_{5}+\lambda_{6}$, respectively.

An early realization in terms of three-dimensional (quantum) Hilbert space can, for instance, be found in Ref. [35] (pp. 5392, 5393); other such parametrizations are discussed in Refs. [88,130,131,132].

The full hull problem, including all joint expectations of dichotomic ±1 observables yields 64 inequalities. The full hull computations for the probabilities $p_{1},\ldots ,p_{10}$ on all atoms $a_{1},\ldots ,a_{10}$ reduces to 16 inequalities. If one considers only the five probabilities on the intertwining atoms, then the Bub-Stairs inequalit $p_{1}+p_{3}+p_{5}+p_{7}+p_{9}\leq 2$ results [130,131,132]. Concentration on the four non-intertwining atoms yields $p_{2}+p_{4}+p_{6}+p_{8}+p_{10}\geq 1$. Limiting the hull computation to adjacent pair expectations of dichotomic ±1 observables yields the Klyachko-Can-Biniciogolu-Shumovsky inequality [88].

#### 5.5.3. Deterministic Predictions and Probabilities on the Specker Bug

Next, we shall study a quasiclassical collection of observables with the property that the preparation of the system in a particular state entails the prediction that another particular observable must have a particular null/zero outcome. The respective quantum observables violate that classical constraint by predicting a non-zero probability of the outcome. Therefore, by observing an outcome one could “certify” non-classicality relative to the (idealistic) assumptions that this particular quasiclassical collection of counterfactual observables “exists” and has relevance to the situation.

History: The pasting of two house/pentagon/pentagram logics with three common contexts results in ever tighter conditions for two-valued measures and thus truth-value assignments: consider the Greechie orthogonality hypergraph of a logic drawn in Figure 1d. Specker [133] called this the “Käfer” (German for bug) because its graph remotely (for Specker) resembled the shape of a bug. In 1965 it was introduced by Kochen and Specker (Figure 1, p. 182 in [134]) who subsequently used it as a subgraph in the diagrams $\Gamma_{1}$, $\Gamma_{2}$ and $\Gamma_{3}$ demonstrating the existence of quantum propositional structures with the “true implies true” property (cf. Section 5.5.4), the non-existence of any two-valued state (cf. Section 5.7), and the existence of a non-separating set of two-valued states (cf. Section 5.6.1), respectively [122].

Pitowsky called this gadget “cat’s cradle” [135,136]. See also Figure 1, p. 123 in [137] (reprinted in Ref. [138]), a subgraph in Figure 21, pp. 126–127 in [139], Figure B.l. p. 64 in [140], pp. 588–589 in [141], Figure 2, p. 446 in [142], and Figure 2.4.6, p. 39 in [39] for early discussions.

The Specker bug/cat’s cradle logic is a pasting of seven intertwined contexts $\{\left\{a_{1},a_{2},a_{3}\right\}$, $\left\{a_{3},a_{4},a_{5}\right\}$, $\left\{a_{5},a_{6},a_{7}\right\}$, $\left\{a_{7},a_{8},a_{9}\right\}$, $\left\{a_{9},a_{10},a_{11}\right\}$, $\left\{a_{11},a_{12},a_{1}\right\}$, $\left\{a_{4},a_{13},a_{10}\right\}\}$ from two houses/pentagons/pentagrams $\{\left\{a_{1},a_{2},a_{3}\right\}$, $\left\{a_{3},a_{4},a_{5}\right\}$, $\left\{a_{9},a_{10},a_{11}\right\}$, $\left\{a_{11},a_{12},a_{1}\right\}$, $\left\{a_{4},a_{13},a_{10}\right\}\}$ and $\{\left\{a_{3},a_{4},a_{5}\right\}$, $\left\{a_{5},a_{6},a_{7}\right\}$, $\left\{a_{7},a_{8},a_{9}\right\}$, $\left\{a_{9},a_{10},a_{11}\right\}$, $\left\{a_{4},a_{13},a_{10}\right\}\}$ with three common blocks $\left\{a_{3},a_{4},a_{5}\right\}$, $\left\{a_{9},a_{10},a_{11}\right\}$, and $\left\{a_{4},a_{13},a_{10}\right\}$. The Greechie orthogonality hypergraph of the Specker bug/cat’s cradle gadget is depicted in Figure 1d.

Classical interpretations: The Specker bug/cat’s cradle logic allows 14 two-valued states which are listed in Table 4.

An early realization in terms of partition logics can be found in Refs. [27,35]. An explicit faithful orthogonal realization in $R^{3}$ consisting of 13 suitably chosen projections (p. 206, Figure 1 in [143]) (see also (Figure 4, p. 5387 in [35])). It is not too difficult to read the canonical partition logic (up to permutations) off the set of all 14 two-valued states which are tabulated in Table 4:(15)$$\begin{array}{c} \{\{\{1,2,3\},\{4,5,6,7,8,9\},\{10,11,12,13,14\}\},\\ \{\{10,11,12,13,14\},\{2,6,7,8\},\{1,3,4,5,9\}\},\\ \{\{1,3,4,5,9\},\{2,6,8,11,12,14\},\{7,10,13\}\},\\ \{\{7,10,13\},\{3,5,8,9,11,14\},\{1,2,4,6,12\}\},\\ \{\{1,2,4,6,12\},\{3,9,13,14\},\{5,7,8,10,11\}\},\\ \{\{5,7,8,10,11\},\{4,6,9,12,13,14\},\{1,2,3\}\},\\ \{\{2,6,7,8\},\{1,4,5,10,11,12\},\{3,9,13,14\}\}\}. \end{array}$$

Deterministic prediction: As pointed out by Greechie (Figure 1, p. 122–123 in [137] and reprinted in Ref. [138]), Pták and Pulmannová (Figure 2.4.6, p. 39 in [39]) as well as Pitowsky [135,136] the reduction of some probabilities of atoms at intertwined contexts yields (p. 285, Equation (11.2) in [91])
(16)$$p_{1}+p_{7}=\frac{3}{2}-\frac{1}{2}\left(p_{12}+p_{13}+p_{2}+p_{6}+p_{8}\right)\leq \frac{3}{2}.$$

A tighter approximation comes from the explicit parameterization of the classical probabilities on the atoms $a_{1}$ and $a_{7}$, derivable from all the mutually disjoined two-valued states which do not vanish on those atoms (Figure 12.9 in [108]): $p_{1}=\lambda_{1}+\lambda_{2}+\lambda_{3}$, and $p_{7}=\lambda_{7}+\lambda_{10}+\lambda_{13}$. Because of additivity the 14 positive weights $\lambda_{1},\ldots ,\lambda_{14}\geq 0$ must add up to 1; that is, $\sum_{i=1}^{14}\lambda_{i}=1$. Therefore,
(17)$$\begin{array}{c} p_{1}+p_{7}=\lambda_{1}+\lambda_{2}+\lambda_{3}+\lambda_{7}+\lambda_{10}+\lambda_{13}\leq \sum_{i=1}^{14}\lambda_{i}=1. \end{array}$$

For two-valued measures this yields the “1-0” or “true implies false” rule [144]: if $a_{1}$ is true, then $a_{7}$ must be false. This property has been exploited in Kochen and Specker’s graph $\Gamma_{1}$ in [122] implementing a “1-1” or “true implies true” rule, as well as to construct both a $\Gamma_{3}$-logic with a non-separating, as well as $\Gamma_{2}$ which does not support a two-valued state. The former “1-1” or “true implies true” case will be discussed in the next section.

Probabilistic prediction: The hull problem yields 23 facet inequalities; one of them relating $p_{1}$ to $p_{7}$: $p_{1}+p_{2}+p_{7}+p_{6}\geq 1+p_{4}$, which is satisfied, since, by subadditivity, $p_{1}+p_{2}=1-p_{3}$, $p_{7}+p_{6}=1-p_{5}$, and $p_{4}=1-p_{5}-p_{3}$. A restricted hull calculation for the joint expectations on the six edges of the Greechie orthogonality hypergraph yields 18 inequalities; among them
(18)$$E_{13}+E_{57}+E_{9,11}\leq E_{35}+E_{79}+E_{11,1}.$$

Configurations in arbitrary dimensions greater than two can be found in Ref. [145], where also another interesting “true-implies-triple false” structure by Yu and Oh (Figure 2 in [44]) is reviewed. Geometric constraints do not allow faithful orthogonal representations of the Specker bug logic which “perform better” than with probability $\frac{1}{9}$ [35,139,140,146,147], so that quantum systems prepared in a pure state corresponding to $a_{1}$ are measured in another pure state $a_{7}$ with a probability higher than $\frac{1}{9}$. More recently “true implies false” gadgets [43,119,121] allow “tunable” relative angles between preparation and measurement states, so that the relative quantum advantage can be made higher. In particular, Ref. [119] contains a collection of 34 observables in 21 intertwined contexts $\left\{a_{3},a_{7},a_{6}\right\}$, $\left\{a_{3},a_{14},a_{15}\right\}$, $\left\{a_{3},a_{10},a_{11}\right\}$, $\left\{a_{8},a_{9},a_{20}\right\}$, $\left\{a_{4},a_{18},a_{21}\right\}$, $\left\{a_{16},a_{17},a_{20}\right\}$, $\left\{a_{12},a_{13},a_{20}\right\}$, $\left\{a_{2},a_{5},a_{19}\right\}$, $\left\{a_{7},a_{24},a_{8}\right\}$, $\left\{a_{5},a_{6},a_{23}\right\}$, $\left\{a_{3},c,a_{1}\right\}$, $\left\{a_{10},a_{34},a_{9}\right\}$, $\left\{a_{2},a_{29},a_{1}\right\}$, $\left\{a_{19},a_{30},a_{22}\right\}$, $\left\{a_{1},a_{32},a_{4}\right\}$, $\left\{a_{22},a_{33},a_{21}\right\}$, $\left\{a_{20},c,a_{22}\right\}$, $\left\{a_{14},a_{25},a_{13}\right\}$, $\left\{a_{15},a_{26},a_{16}\right\}$, $\left\{a_{11},a_{27},a_{12}\right\}$, $\left\{a_{17},a_{31},a_{18}\right\}$. Its Greechie orthogonality hypergraph is drawn in Figure 2. The set of 89 2-valued states induce a “true-implies-false” property on the two pairs $a_{1}$-$a_{20}$ and $a_{1}$-$a_{22}$, respectively.

#### 5.5.4. Deterministic Predictions of Kochen-Specker’s $\Gamma_{1}$ “True Implies True” Logic

Here we shall study a quasiclassical collection of observables with the property that the preparation of the system in a particular state entails the prediction that the outcome of another particular observable must occur with certainty; that is, this outcome must always occur. The respective quantum observables violate that classical constraint by predicting a probability of outcome strictly smaller than one. Therefore, by observing the absence of an outcome one could “certify” non-classicality relative to the (idealistic) assumptions that this particular quasiclassical collection of counterfactual observables “exists” and has relevance to the situation.

Scheme: As depicted in Figure 3 a scheme involving three cyclically arranged three element contexts can serve as a “true implies true” gadget. This triangular scheme has to be adopted by substituting a Specker bug/cat’s cradle in order to fulfill faithful orthogonal representability.

History: A small extension of the Specker bug logic by two contexts extending from $a_{1}$ and $a_{7}$, both intertwining at a point *c* renders a logic which facilitates that whenever $a_{1}$ is true, so must be an atom $b_{1}$, which is element in the context $\left\{a_{7},c,b_{1}\right\}$. The Greechie orthogonality hypergraph of the extended Specker bug (Kochen-Specker’s $\Gamma_{1}$ ([122] (p. 68))) gadget is depicted in Figure 1e.

Other “true implies true” logics were introduced by Belinfante (Figure C.l. p. 67 in [140]), Pitowsky [149] (p. 394), Clifton [142,150,151], as well as Cabello and G. García-Alcaine (Lemma 1 in [152]). More recently “true implies true” gadgets allowing arbitrarily small relative angles between preparation and measurement states, so that the relative quantum advantage can be made arbitrarily high [43,121]. Configurations in arbitrary dimensions greater than two can be found in Ref. [145].

Classical interpretations: The reduction of some probabilities of atoms at intertwined contexts yields ($q_{1},q_{7}$ are the probabilities on $b_{1},b_{7}$, respectively), additionally to Equation (Equation 16), $p_{1}-p_{7}=q_{1}-q_{7}$. This implies that for all the 112 two-valued states, if $p_{1}=1$, then [from Equation (Equation 16)] $p_{7}=0$, and $q_{1}=1$ as well as $q_{7}=1-q_{1}=0$.

### 5.6. Beyond Classical Embedability

The following examples of observables are situated in-between classical (structure-preserving) faithful embeddability into Boolean algebras on the one hand, and a total absence of any classical valuations on the other hand. Kochen-Specker showed that this kind of classical embeddability of a (quantum) logic can be characterized by a separability criterion (Theorem 0 in [122]) related to its set of two-valued states: a logic has a separable set of two-valued states if any arbitrarily chosen pair $a_{i}$, $a_{j}$, i≠j, of different atoms/elementary observable propositions of that logic can be “separated” by at least one of those states, say *v* such that “*v* discriminates between $a_{i}$ and $a_{j}$”; that is, $v\left(a_{i}\right)\neq v\left(a_{j}\right)$.

#### 5.6.1. Deterministic Predictions on a Combo of Two Interlinked Specker Bugs

The following collection of observables allows a faithful orthogonal representation (in three- and higher-dimensional Hilbert spaces) and thus a quantum interpretation. However, it does not allow a set of 24 two-valued states, enumerated in Table 5 which empirically separates any observable from any other: some observables always have the same classical value, and therefore (unlike quantum observables) cannot be differentiated by any classical means.

Scheme: A “bowtie” scheme depicted in Figure 4 serves as a straightforward implementation of a logic with an non-separating set of two-valued states

History: As we are heading toward logics with less and less “rich” set of two-valued states we are approaching a logic depicted in Figure 1f which is a combination of two Specker bug logics linked by two external contexts. It is the $\Gamma_{3}$-configuration of Kochen-Specker [122] (p. 70) with a set of two-valued states which is no longer separating: In this case, one obtains the “one-one” and “zero-zero rules” [144], stating that $a_{1}$ occurs if and only if $b_{1}$ occurs (likewise, $a_{7}$ occurs if and only if $b_{7}$ occurs): Suppose *v* is a two-valued state on the $\Gamma_{3}$-configuration of Kochen-Specker. Whenever $v(a_{1})=1$, then v(c)=0 because it is in the same context $\left\{a_{1},c,b_{7}\right\}$ as $a_{1}$. Furthermore, because of Equation (Equation 16), whenever $v(a_{1})=1$, then $v(a_{7})=0$. Because $b_{1}$ is in the same context $\left\{a_{7},c,b_{1}\right\}$ as $a_{7}$ and *c*, because of admissibility, $v(b_{1})=1$. Conversely, by symmetry, whenever $v(b_{1})=1$, so must be $v(a_{1})=1$. Therefore it can never happen that either one of the two atoms $a_{1}$ and $b_{1}$ have different dichotomic values. The same is true for the pair of atoms $a_{7}$ and $b_{7}$.

If oone ties together two Specker bug logics (at their “true implies false” extremities) one obtains non-separability; just extending one to the Kochen-Specker $\Gamma_{1}$ logic [122] (p. 68) discussed earlier to obtain “true implies true” would be insufficient. Because in this case a consistent two-valued state exists for which $v\left(b_{1}\right)=v\left(b_{7}\right)=1$ and $v\left(a_{1}\right)=v\left(a_{7}\right)=0$, thereby separating $a_{1}$ from $b_{1}$, and *vice versa*. A second Specker bug logic is neded to elimitate this case; in particular, $v\left(b_{1}\right)=v\left(b_{7}\right)=1$. The Greechie orthogonality hypergraph of this extended combo of two Specker bug (Kochen-Specker’s $\Gamma_{3}$) logic is depicted in Figure 1f.

Besides the quantum mechanical realization of this logic in terms of propositions which are projection operators corresponding to vectors in three-dimensional Hilbert space suggested by Kochen and Specker [122], Tkadlec has given (p. 206, Figure 1 in [143]) an explicit collection of such vectors (see also the proof of Proposition 7.2 in Ref. [35] (p. 5392)).

Seizing the classical remainder of this logic results in the “folding” or “merging” of the non-separable observables $a_{1}$-$b_{1}$ as well as $a_{7}$-$b_{7}$ by identifying those pairs, respectively. As a result also the two contexts $\left\{a_{1},c,b_{7}\right\}$ as well as $\left\{a_{7},c,b_{1}\right\}$ merge and fold into each other, leaving a structure depicted in Figure 5. Formally, such a structure is obtained in two steps: (i) by enumerating all two-valued states on the original (non-classical) logic; followed by (ii) an inverse reconstruction of the classical partition logic resulting from the set of two-valued states obtained in step (i).

Embeddability: As every algebra embeddable in a Boolean algebra must have a separating set of two-valued states (Theorem 0 in [122]), this logic is no longer “classical” in the sense of “homomorphically (structure-preserving) imbeddable”. Nevertheless, there may still exist “plenty” of two-valued states–indeed, of them. It is just that these states can no longer differentiate between the pairs of atoms $\left(a_{1},b_{1}\right)$ as well as $\left(a_{7},b_{7}\right)$. Partition logics and their generalized urn or finite automata models fail to reproduce two linked Specker bug logics resulting in a Kochen-Specker $\Gamma_{3}$ logic even at this stage. Of course, the situation will become more dramatic with the non-existence of any kind of two-valued state (interpretable as truth assignment) on certain logics associated with quantum propositions.

Chromatic inseparability: The “true implies true” rule is associated with chromatic separability; in particular, with the impossibility to separate two atoms $a_{7}$ and $b_{7}$ with less than four colors. A proof is presented in (Figure 12.4 in [108]). That chromatic separability on the unit sphere requires 4 colors is implicit in Refs. [153,154].

#### 5.6.2. Deterministic Predictions on Observables with a Nun-Unital Set Of Two-Valued States

In our “escalation of non-classicality”, we “scratch the current borderline” between non-classical observables allowing a faithful orthogonal (and thus quantum mechanical) representation with a very “meager” set of two-valued states: this is “as bad as it might get” before the complete absence of all two-valued states interpretable as classical truth assignments (subject to admissibility rules) Indeed, these states are so scarce that they do not acquire the vale “1” in at least one (but usually many) of its atomic propositions: every classical representation requires these observables not to occur at all times; regardless of the preparation state. These sets of states are called unital [129,143,155].

Schemes and historic examples: Figure 6a–d present hypergraphs of logics with a non-unital set of two-valued states. Whereas the propositional structures depicted in Figure 6a,c have no faithful orthogonal representation in three-dimensional Hilbert space (they contain disallowed “triangles), logics schematically depicted in Figure 6b,d may have a faithful orthogonal representation if the true-implies-false gadget they contain are suitable. Based on a (non-intuitive) configuration (invented for the rendition of a classical tautology which is no quantum tautology) by Schütte [156] and mentioned in a dissertation by Clavadetscher-Seeberger [157] a concrete example of a logic with a non-unital set of two-valued states which has a faithful orthogonal representation in three-dimensional Hilbert space was given by Tkadlec (Figure 2, p. 207 in [143]). It consists of 36 atoms in 26 intertwined contexts; in short $\{\left\{a_{1},a_{2},a_{3}\right\}$, $\left\{a_{1},a_{4},a_{5}\right\}$, $\left\{a_{4},a_{6},a_{7}\right\}$, $\left\{a_{8},a_{9},a_{31}\right\}$, $\left\{a_{7},a_{8},a_{23}\right\}$, $\left\{a_{17},a_{9},a_{2}\right\}$, $\left\{a_{7},a_{33},a_{10}\right\}$, $\left\{a_{28},a_{17},a_{18}\right\}$, $\left\{a_{5},a_{18},a_{19}\right\}$, $\left\{a_{19},a_{20},a_{31}\right\}$, $\left\{a_{19},a_{21},a_{23}\right\}$, $\left\{a_{4},a_{14},a_{15}\right\}$, $\left\{a_{15},a_{20},a_{26}\right\}$, $\left\{a_{14},a_{17},a_{35}\right\}$, $\left\{a_{13},a_{15},a_{16}\right\}$, $\left\{a_{16},a_{22},a_{36}\right\}$, $\left\{a_{21},a_{22},a_{27}\right\}$, $\left\{a_{3},a_{13},a_{23}\right\}$, $\left\{a_{2},a_{22},a_{24}\right\}$, $\left\{a_{11},a_{12},a_{13}\right\}$, $\left\{a_{10},a_{11},a_{34}\right\}$, $\left\{a_{24},a_{25},a_{32}\right\}$, $\left\{a_{5},a_{11},a_{25}\right\}$, $\left\{a_{9},a_{12},a_{29}\right\}$, $\left\{a_{6},a_{24},a_{30}\right\}$, $\left\{a_{2},a_{10},a_{20}\}\right\}$ allowing only six two-valued states enumerated in Table 6.

A closer inspection of observable propositions whosecolumn entries are all “0” reveals that all those classical cases require no less than eight such observable propositions $a_{1}$, $a_{3}$, $a_{9}$, $a_{10}$, $a_{17}$, $a_{20}$, $a_{22}$, $a_{24}$ to be false. On the other hand, all classical two-valued states require the observable $a_{2}$ to “always happen”.

On the other hand, there exists a faithful orthogonal (quantum) representation (Figure 2, p. 207 in [143]) of the Tkadlec logic for which not all of those atoms are mutually collinear or orthogonal. That is, regardless of the state prepared, some of the respective quantum outcomes are sometimes “seen”. With regards to the quantum observable corresponding to $a_{2}$ one might choose a quantum state perpendicular to the unit vector representing $a_{2}$, and thereby arrive at a complete contradiction to the classical prediction.

#### 5.6.3. Direct Probabilistic Criteria against Value Definiteness from Constraints on Two-Valued Measures

The “1-1” or “true implies true” rule can be taken as an operational criterion: Suppose that one prepares a system to be in a pure state corresponding to $a_{1}$, such that the preparation ensures that $v(a_{1})=1$. If the system is then measured along $b_{1}$, and the proposition that the system is in state $b_{1}$ is found to be *not* true, meaning that $v(b_{1})\neq 1$ (the respective detector does not click), then one has established that the system is not performing classically, because classically the set of two-valued states requires non-separability; that is, $v\left(a_{1}\right)=v\left(b_{1}\right)=1$. With the Tkadlec directions (p. 206, Figure 1 in [143]) (see also (Figure 4, p. 5387 in [35])), $|a_{1}\rangle =\left(1/\sqrt{3}\right){\left(1,\sqrt{2},0\right)}^{⊺}$ and $|b_{1}\rangle =\left(1/\sqrt{3}\right){\left(\sqrt{2},1,0\right)}^{⊺}$ so that the probability to find a quantized system prepared along $|a_{1}\rangle$ and measured along $|b_{1}\rangle$ is $p_{a_{1}}\left(b_{1}\right)={\left|\langle b_{1}|a_{1}\rangle \right|}^{2}=8/9$, and that a violation of classicality should occur with “optimal” [35,146,147] (for any fathful orthogonal representation) probability 1/9. Of course, any other classical prediction, such as the “1-0” or “true implies false” rule, or more general classical predictions such as of Equation (Equation 16) can also be taken as empirical criteria for non-classicality (Section 11.3.2. in [91])).

Indeed, already Stairs [141] (pp. 588–589) has argued along similar lines for the Specker bug “true implies false” logic (a translation into our nomenclature is: $m1\left(1\right)\equiv a_{1}$, $m2\left(1\right)\equiv a_{3}$, $m2\left(2\right)\equiv a_{5}$, $m2\left(3\right)\equiv a_{4}$, $m3\left(1\right)\equiv a_{11}$, $m3\left(2\right)\equiv a_{9}$, $m3\left(3\right)\equiv a_{10}$, $m4\left(1\right)\equiv a_{7}$). Independently Clifton (there is a note added in proof to Stairs [141] (pp. 588–589)) presents asimilar argument, based on (i) another “true implies true” logic (Sections II and III, Figure 1 in [142,150,151]) inspired by Bell (Figure C.l. p. 67 in [140]) (cf. also Pitowsky [149] (p. 394)), as well as (ii) on the Specker bug logic (Section IV, Figure 2 in [142]). More recently Hardy [158,159,160] as well as Cabello and García-Alcaine and others [132,161,162,163,164,165] discussed such scenarios. These criteria for non-classicality are benchmarks aside from the Boole-Bell type polytope method, and also different from the full Kochen-Specker theorem.

Any such criteria can be directly applied also to logics with a non-unital set of two-valued states. In such cases, the situation can be even more stringent because any classical prediction “locks” the observables into non-occurrence. However, when interpreting those configurations of observables quantum mechanically – that is, as a faithful orthogonal representation – this can never be uniformly achieved: because, if there is only one observable forced into being “silent” then one can always choose a preparation state which is non-orthogonal to the one predicted to be “silent”. If there are more “silent” and complementary (non-collinear and non-orthogonal) observables such as in Tkadlec’s logic (Figure 2, p. 207 in [143]) any attempt to accommodate the quantum predictions with the classical “silent” ones are doomed from the very beginning: one just needs to be waiting for the first click in a detector corresponding to one of the classically “silent” observables to be able to claim the non-classicality of quantum mechanics.

Very similar issues relating to chromatic separability and embeddability as have been put forward in the case of non-separating sets of two-valued states can be made for cases on non-unital sets of two-valued states.

### 5.7. Finite Propositional Structures Admitting Neither Truth Assignments nor Predictions

#### 5.7.1. Scarcity of Two-Valued States

When it comes to the absence of a global two-valued state on quantum logics corresponding to Hilbert spaces of dimension three and higher–whenever contexts or blocks can be intertwined or pasted [116] to form chains–Kochen and Specker [122] pursued a very concrete, “constructive” (in the sense of finitary mathematical objects but not in the sense of physical operationalizability [166]) strategy: they presented finite logics realizable by vectors (from the origin to the unit sphere) spanning one-dimensional subspaces, equivalent to observable propositions, which allowed for lesser & lesser two-valued state properties.

History: For non-homomorphic imbedability (Theorem 0 in [122]) it is already sufficient to present finite collections of observables with a non-separating or non-unital set of two-valued states. Concrete examples have already been exposed by considering the Specker bug combo $\Gamma_{3}$ [122] (p. 70) discussed in Section 5.6.1, and the Tkadlec non-unital logic based on the Schütte rays discussed in Section 5.6.2, respectively.

Kochen and Specker went further and presented a proof by contradiction of the non-existence of two-valued states on a finite number of propositions, based on their $\Gamma_{1}$ “true implies true” logic [122] (p. 68) discussed in Section 5.5.4, and iterating them until they reached a complete contradiction in their $\Gamma_{2}$ logic [122] (p. 69). As was pointed out earlier, their representation as points of the sphere is a little bit “buggy” (as could be expected from the formation of so many bugs): as Tkadlec has observed, Kochen-Specker diagram $\Gamma_{2}$ it is not a one-to-one representation of the logic, because of some different points at the diagram represent the same element of corresponding orthomodular poset [cf. Ref. [35] (p. 5390), and Ref. [167] (p. 156)].

The early 1990s saw an ongoing flurry of papers recasting the Kochen-Specker proof with ever smaller numbers of, or more symmetric, configurations of observables (see Refs. [35,44,143,162,167,168,169,170,171,172,173,174,175,176,177,178,179,180,181,182,183,184,185,186] for an incomplete list). The “most compact” proofs (in terms of the number of vectors and their associated observables) should contain no less than 22 vectors in three-dimensional space [187], and no less than 18 vectors for dimension four [179] and higher [180,188]. In four dimensions the most compact explicit realization has been suggested by Cabello, Estebaranz and García-Alcaine [162,179,189]. It consists of 9 contexts, with each of the 18 atoms tightly intertwined in two contexts. The challenge in such (mostly automated) computations is twofold: to generate (and exclude a sufficient number) of “candidate graphs”; and subsequently to find a faithful orthogonal representation [124,125,126,185,186].

#### 5.7.2. Chromatic Number of the Sphere

Graph coloring allows another view on value (in)definiteness. The chromatic number of a graph is defined as the least number of colors needed in any total coloring of a graph; with the constraint that two adjacent vertices have distinct colors.

Suppose that we are interested in the chromatic number of graphs associated with both (i) the real and (ii) the rational three-dimensional unit sphere.

More generally, we can consider *n*-dimensional unit spheres with the same adjacency property defined by orthogonality. An orthonormal basis will be called context (block, maximal observable, Boolean subalgebra), or, in this particular area, a *n*-clique. Note that for any such graphs involving *n*-cliques the chromatic number of this graph is at least be *n* (because the chromatic number of a single *n*-clique or context is *n*).

Therefore vertices of the graph are identified with points on the three-dimensional unit sphere; with adjacency defined by orthogonality; that is, two vertices of the graph are adjacent if and only if the unit vectors from the origin to the respective two points are orthogonal.

The connection to quantum logic is this: any context (block, maximal observable, Boolean subalgebra, orthonormal basis) can be represented by a triple of points on the sphere such that any two unit vectors from the origin to two distinct points of that triple of points are orthogonal. Thus graph adjacency in logical terms indicates “belonging to some common context (block, maximal observable, Boolean subalgebra, orthonormal basis)”.

In three dimensions, if the chromatic number of graphs is four or higher, there does not globally exist any consistent coloring obeying the rule that adjacent vertices (orthogonal vectors) must have different colors: if one allows only three different colors, then somewhere in that graph of a chromatic number higher than three, adjacent vertices must have the same colors (or else the chromatic number would be three or lower).

By a similar argument, non-separability of two-valued states – such as encountered in Section 5.6.1 with the $\Gamma_{3}$-configuration of Kochen-Specker [122] (p. 70)–translates into non-separability by colorings with colors less or equal to the number of atoms in a block [cf. Figure 1f].

Godsil and Zaks [153,154] proved the following results relevant to this discussion:(i)the chromatic number of the graph based on points of real-valued unit 2-sphere $S^{2}$ is four (Lemma 1.1 in [153]).(ii)The chromatic number of rational points on the unit 2-sphere $S^{2}\cap Q^{3}$ is three (Lemma 1.2 in [153]).

We shall concentrate on (i) and discuss (ii) later. As was pointed out by Godsil in an email conversation from 13 March 2016 [190], *“the fact that the chromatic number of the unit sphere in $R^{3}$ is four is a consequence of Gleason’s theorem, from which the Kochen-Specker theorem follows by compactness. Gleason’s result implies that there is no subset of the sphere that contains exactly one point from each orthonormal basis”.*

Indeed, any coloring can be mapped onto a two-valued state by identifying a single color with “1” and all other colors with “0”. By reduction, all propositions on two-valued states translate into statements about graph coloring. In particular, if the chromatic number of any logical structure representable as a graph consisting of *n*-atomic contexts (blocks, maximal observables with *n* outcomes, Boolean subalgebras $2^{n}$, orthonormal bases with *n* elements)–for instance, as Greechie orthogonality hypergraph of quantum logics–is larger than *n*, then there cannot be any globally consistent two-valued state (truth-value assignment) obeying adjacency (aka admissibility). Likewise, if no two-valued states on a logic which is a pasting of *n*-atomic contexts exist, then, by reduction, no global consistent coloring with *n* different colors exists. Therefore, the Kochen-Specker theorem proves that the chromatic number of the graph corresponding to the unit sphere with adjacency defined as orthogonality must be higher than three.

For an inverse construction, one may conjecture that all colorings of a particular Greechie orthogonal hypergraph are determined by the set of unital two-valued states (if it exists) (modulo permutations of colors). This can be made plausible by the following construction: Choose one context or block and choose the first atom thereof. Then take one of the two-valued states which acquire the value “1” on this atom, and associate the first color with this state. Accordingly, all atoms whose value is “1” in this aforementioned two-valued state become colored with this particular color as well in all the other (complementary) contexts or blocks. Then take the second atom of the original block and repeat this procedure with a second color, and so on until the last atom is reached. All other colorings can be obtained by all variations of two-valued states, respectively.

Based on Godsil and Zaks finding that the chromatic number of rational points on the unit sphere $S^{2}\cap Q^{3}$ is three (Lemma 1.2 in [153])—thereby constructing a two-valued measure on the rational unit sphere by identifying one color with “1” and the two remaining colors with “0”—there exist “exotic” options to circumvent Kochen-Specker type constructions which were quite aggressively (Cabello has referred to this as the second contextuality war [191]) marketed by allegedly “nullifying” [192] the respective theorems under the umbrella of “finite precision measurements” [193,194,195,196,197,198]: the support of vectors spanning the one-dimensional subspaces associated with atomic propositions could be “diluted” yet dense, so much so that the intertwines of contexts (blocks, maximal observables, Boolean subalgebras, orthonormal bases) break up; and the contexts themselves become “free and isolated”. Under such circumstances the logics decay into horizontal sums; and the Greechie orthogonality hypergraphs are just disconnected stacks of previously intertwined contexts. As can be expected, proofs of Gleason- or Kochen-Specker-type theorems do no longer exist, as the necessary intertwines are missing.

The “nullification” claim and subsequent ones triggered a lot of papers, some cited in [198]; mostly critical—of course, not of the results of Godsil and Zaks’s finding; how could they?—but with respect to their physical applicability. Peres even wrote a parody by arguing that “finite precision measurement nullifies Euclid’s postulates” [199], so that “nullification” of the Kochen-Specker theorem might have to be our least concern.

#### 5.7.3. Exploring Value Indefiniteness

“Extensions” of the Kochen-Specker theorem investigate situations in which a system is prepared in a state |x〉〈x| along direction |x〉 and measured along a non-orthogonal, non-collinear projection |y〉〈y| along direction |y〉. Those extensions yield what may be called [120,200] *indeterminacy*: Pitowsky’s *logical indeterminacy principle* (Theorem 6, p. 226 in [120]) states that given two linearly independent non-orthogonal unit vectors |x〉 and |y〉 in $R^{3}$, there is a finite set of unit vectors Γ(|x〉,|y〉) containing |x〉 and |y〉 for which the following statements hold: There is no two-valued state *v* on Γ(|x〉,|y〉) which satisfies
(i)either v(|x〉)=v(|y〉)=1,(ii)or v(|x〉)=1 and v(|y〉)=0,(iii)or v(|x〉)=0 and v(|y〉)=1.

Stated differently (Theorem 2, p. 183 in [200]), let |x〉 and |y〉 be two non-orthogonal rays in a Hilbert space H of finite dimension ≥3. Then there is a finite set of rays Γ(|x〉,|y〉) containing |x〉 and |y〉 such that a two-valued state *v* on Γ(|x〉,|y〉) satisfies v(|x〉),(|y〉)∈{0,1} only if v(|x〉)=v(|y〉)=0. That is, if a system of three mutually exclusive outcomes (such as the spin of a spin-1 particle in a particular direction) is prepared in a definite state |x〉 corresponding to v(|x〉)=1, then the state v(|y〉) along some direction |y〉 which is neither collinear nor orthogonal to |x〉 cannot be (pre-)determined, because, by an argument *via* some set of intertwined rays Γ(|x〉,|y〉), both cases would lead to a complete contradiction.

The proofs of the logical indeterminacy principle presented by Pitowsky and Hrushovski [120,200] is global in the sense that any ray in the set of intertwining rays Γ(|x〉,|y〉) in-between |x〉 and |y〉—and thus not necessarily the “beginning and end points” |x〉 and |y〉–may not have a pre-existing value. (If you are an omni-realist, substitute “pre-existing” by “non-contextual”: that is, any ray in the set of intertwining rays Γ(|x〉,|y〉) may violate the admissibility rules and, in particular, non-contextuality.) Therefore, one might argue that the cases (i) as well as (ii); that is, v(|x〉)=v(|y〉)=1. as well as v(|x〉)=1 and v(|y〉)=0 might still be predefined, whereas at least one ray in Γ(|x〉,|y〉) cannot be pre-defined. (If you are an omni-realist, substitute “pre-defined” with “non-contextual”).

This possibility was excluded in a series of papers [40,41,42,43] localizing value indefiniteness. Therefore, the strong admissibility rules coinciding with two-valued states which are total function on a logic, were generalized or extended (if you prefer “weakened”) to allow value indefiniteness. Essentially, by allowing the two-valued state to be a partial function on the logic, which need not be defined any longer on all of its elements, admissibility was defined by the rules WAD1-WAD3 of Section 5.1, as well as counterfactually, in all contexts including |x〉 and in mutually orthogonal rays which are orthogonal to |x〉, such as the star-shaped Greechie orthogonal hypergraph configuration (Figure 5 in [42]) (see also Figure 1 of Ref. [201]).

In such a formalism, and relative to the assumptions—in particular, by the admissibility rules WAD1-WAD3 allowing for value indefiniteness, sets of intertwined rays Γ(|x〉,|y〉) can be constructed which render value indefiniteness of property |y〉〈y| if the system is prepared in state |x〉 (and thus v(|x〉)=1). More specifically, finite sets of intertwined rays Γ(|x〉,|y〉) can be found which demonstrate that in accordance with the “weak” admissibility rules WAD1-WAD3 of Section 5.1, in Hilbert spaces of dimension greater than two, in accordance with complementarity, any proposition which is complementary with respect to the state prepared must be value indefinite [40,41,42,43].

#### 5.7.4. How Can You Measure a Contradiction?

Clifton replied with this (rhetorical) question after I had asked him if he could imagine any possibility to somehow “operationalize” the Kochen-Specker theorem. Indeed, the Kochen-Specker theorem—in particular, not only non-separability but the total absence of any two-valued state—was resilient to attempts to somehow “measure” it: first, as alluded by Clifton, its proof is by contraction—any assumption or attempt to consistently (by admissibility) construct a two-valued state on certain finite subsets of quantum logic provably fails.

Second, the very absence of any two-valued state on such logics reveals the futility of any attempt to somehow define classical probabilities or classical predictions; let alone the derivation of any Boole’s conditions of physical experience—both rely on, or are, the hull spanned by the vertices derivable from two-valued states (if the latter existed) and the respective correlations. Therefore, in essence, on logics corresponding to Kochen-Specker configurations, such as the $\Gamma_{2}$-configuration of Kochen-Specker [122] (p. 69), or the Cabello, Estebaranz, and García-Alcaine logic [162,189] which (subject to admissibility) have no two-valued states, classical probability theory breaks down entirely—that is, in the most fundamental way; by not allowing any two-valued state.

It is amazing how many papers exist which claim to “experimentally verify” the Kochen-Specker theorem. However, without exception, those experiments either prove some kind of Bell-Boole of inequality on single-particles (to be fair this is referred to as “proving contextuality”; such as, for instance, Refs. [202,203,204,205,206]); or show that the quantum predictions yield complete contradictions if one “forces” or assumes the counterfactual co-existence of observables in different contexts (and measured in separate, distinct experiments carried out in different subensembles; e.g., Refs. [189,207,208,209]; again these lists of references are incomplete.)

Of course, what one could still do is measuring all contexts, or subsets of compatible observables (possibly by Einstein-Podolsky-Rosen type [70] counterfactual inference)—one at a time—on different subensembles prepared in the same state by Einstein-Podolsky-Rosen type [70] experiments, and comparing the complete sets of results with classical predictions [207].

#### 5.7.5. Non-Contextual Inequalities

If one is willing to drop admissibility altogether while at the same time maintaining non-contextuality—thereby only assuming that the hidden variable theories assign values to all the observables (Section 4, p. 375 in [210]), thereby only assuming non-contextuality [100], one arrives at *non-contextual inequalities* [211]. Of course, these value assignments need to be much more general as the admissibility requirements on two-valued states; allowing all $2^{n}$ (instead of just *n* combinations) of contexts with *n* atoms; such as 1−1−1−…−1, or 0−0−…−0.

## 6. Quantum Predictions: Probabilities and Expectations

Since from Hilbert space dimension higher than two, there do not exist any two-valued states, the (quasi-)classical Boolean strategy to find (or define) probabilities *via* the convex sum of two-valued states brakes down entirely. Therefore, the quantum probabilities have to be “derived” or postulated from entirely new concepts, based on quantities—such as vectors or projection operators—in linear vector spaces equipped with a scalar product [212,213,214,215,216,217]. One implicit property and guiding principle of these “new type of probabilities” was that among those observables which are simultaneously co-measurable (that is, whose projection operators commute), the classical Kolmogorovian-type probability theory should hold.

Historically, what is often referred to as Born rule for calculating probabilities, was a statistical re-interpretation of Schrödinger’s wave function (Footnote 1, Anmerkung bei der Korrektur, [218] (p. 865)), as outlined by Dirac [212,213] (a digression: a small piece [219] on “the futility of war” by the late Dirac is highly recommended; I had the honour listening to the talk personally), Jordan [214], von Neumann [215,216,217], and Lüders [220,221,222].

Rather than stating it as an axiom of quantum mechanics, Gleason [36] derived the Born rule from elementary assumptions; in particular from subclassicality: within contexts—that is, among mutually commuting and thus simultaneously co-measurable observables—the quantum probabilities should reduce to the classical, Kolmogorovian, form. In particular, the probabilities of propositions corresponding to observables which are (i) mutually exclusive (in geometric terms: correspond to orthogonal vectors/projectors) as well as (ii) simultaneously co-measurable observables are (i) non-negative, (ii) normalized, and (iii) finite additive; that is, probabilities (of atoms within contexts or blocks) add up (Section 1 in [223]).

As already mentioned earlier, Gleason’s paper made a high impact on those in the community capable of comprehending it [101,120,122,224,225,226,227,228]. Nevertheless, it might not be unreasonable to state that while a proof of the Kochen-Specker theorem is straightforward, Gleason’s results are less attainable. However, in what follows we shall be less concerned with either necessity nor with mixed states, but shall rather concentrate on sufficiency and pure states. (This will also rid us of the limitations to Hilbert spaces of dimensions higher than two.)

Independently, and presumably motivated from Lovász’s faithful orthogonal representation of graphs by vectors in some Hilbert space [124,125,126], Grötschel, Lovász and Schrijver (Theorem 3.2, p. 338 in [229] and § 9.3, p. 285-303 in [230]) proposed a (probability) weight [231] on a given graph which essentially rephrases the Born rule in a graph theoretical setting [24,188,232].

Recall that pure states [212,213] as well as elementary yes-no propositions [15,216,217] can both be represented by (normalized) vectors in some Hilbert space. If one prepares a pure state corresponding to a unit vector |x〉 (associated with the one-dimensional projection operator $E_{x}=\left|x\rangle \langle x\right|$) and measures an elementary yes-no proposition, representable by a one-dimensional projection operator $E_{y}=\left|y\rangle \langle y\right|$ (associated with the vector |y〉), then Gleason notes [36] (p. 885) in the second paragraph that (in Dirac notation), *“it is easy to see that such a [[probability]] measure μ can be obtained by selecting a vector |y〉 and, for each closed subspace A, taking μ(A) as the square of the norm of the projection of |y〉 on A”.*

Since in Euclidean space, the projection $E_{y}$ of |y〉 on A=span(|x〉) is the dot product (both vectors |x〉,|y〉 are supposed to be normalized) |x〉〈x|y〉=|x〉cos∠(|x〉,|y〉), Gleason’s observation amounts to the well-known quantum mechanical cosine square probability law referring to the probability to find a system prepared a in state in another, observed, state. (Once this is settled, all self-adjoint observables follow by linearity and the spectral theorem.)

In this line of thought, “measurement” contexts (orthonormal bases) allow “views” on “prepared” contexts (orthonormal bases) by the respective projections.

### 6.1. Gleason-Type Continuity

Gleason’s theorem [36] was a response to Mackey’s problem to *“determine all measures on the closed subspaces of a Hilbert space”* contained in a review [233] of Birkhoff and von Neumann’s centennial paper [15] on the logic of quantum mechanics. Starting from von Neumann’s formalization of quantum mechanics [216,217], the quantum mechanical probabilities and expectations (aka the Born rule) are essentially derived from (sub)additivity among the quantum context; that is, from subclassicality: within any context (Boolean subalgebra, block, maximal observable, orthonormal base) the quantum probabilities sum up to 1.

Gleason’s finding caused ripples in the community, at least of those who cared and coped with it [101,120,122,224,225,226,227,228]. (I recall arguing with Van Lambalgen around 1983, who could not believe that anyone in the larger quantum community had not heard of Gleason’s theorem. As we approached an elevator at Vienna University of Technology’s Freihaus building we realized there was also one very prominent member of the Vienna experimental community entering the cabin. I suggested to stage an example by asking; and *voila**…*)

With the possible exception of Specker—who did not explicitly refer to the Gleason’s theorem in independently announcing that two-valued states on quantum logics cannot exist [38]—he preferred to discuss scholastic philosophy; at that time the Swiss may have inhabited their biotope of quantum logical thinking—Gleason’s theorem directly implies the absence of two-valued states. Indeed, at least for finite dimensions [234,235], as Zierler and Schlessinger [224] (even before publication of Bell’s review [101]) noted, “it should also be mentioned that, in fact, the non-existence of two-valued states is an elementary geometric fact contained quite explicitly in (Paragraph 2.8 in [36])”.

Now, Gleason’s Paragraph 2.8 contains the following main (necessity) theorem [36] (p. 888): *“Every non-negative frame function on the unit sphere S in $R^{3}$ is regular”.* Whereby [36] (p. 886) *“a frame function f [[satisfying additivity]] is regular if and only if there exists a self-adjoint operator T defined on [[the separable Hilbert space]] H such that f(|x〉)=〈Tx|x〉 for all unit vectors |x〉”.* (Of course, Gleason did not use the Dirac notation.)

In what follows we shall consider Hilbert spaces of dimension n=3 and higher. Suppose that the quantum system is prepared to be in a pure state associated with the unit vector |x〉, or the projection operator |x〉〈x|.

As all self-adjoint operators have a spectral decomposition [9] (§ 79), and the scalar product is (anti)linear in its arguments, let us, instead of T, only consider one-dimensional orthogonal projection operators $E_{i}^{2}=E_{i}=|y_{i}\rangle \langle y_{i}|$ (formed by the unit vector $|y_{i}\rangle$ which are elements of an orthonormal basis $\{|y_{1}\rangle ,\ldots ,|y_{n}\rangle \}$) occurring in the spectral sum of $T=\sum_{i=1}^{n\geq 3}\lambda_{i}E_{i}$, with $I_{n}=\sum_{i=1}^{n\geq 3}E_{i}$.

Thus if T is restricted to some one-dimensional projection operator E=|y〉〈y| along |y〉, then Gleason’s main theorem states that any frame function reduces to the absolute square of the scalar product; and in real Hilbert space to the square of the angle between those vectors spanning the linear subspaces corresponding to the two projectors involved; that is (note that E is self-adjoint), $f_{y}{(|x\rangle )=\langle Ex|x\rangle =\langle x|Ex\rangle =\langle x|y\rangle \langle y|x\rangle =|\langle x|y\rangle |}^{2}=\cos^{2}∠\left(x,y\right)$.

Hence, unless a configuration of contexts is of the star-shaped Greechie orthogonality hypergraph form as depicted in Figure 5 of Ref. [42] (see also Figure 1 of Ref. [201]),–meaning that they all share one common atom; and, in terms of geometry, meaning that all orthonormal bases share a common vector–and the two-valued state has value 1 on its center, there is no way that any two contexts could have a two-valued assignment; even if one context has one: it is just not possible by the continuous, $\cos^{2}$-form of the quantum probabilities. That is (at least in this author’s believe) the watered-down version of the remark of Zierler and Schlessinger (p. 259, Example 3.2 in [224]).

### 6.2. Comparison of Classical and Quantum form of Correlations

In what follows, quantum configurations corresponding to the logic presented in the earlier sections will be considered. All of them have quantum realizations in terms of vectors spanning one-dimensional subspaces corresponding to the respective one-dimensional projection operators.

It is stated without a detailed derivation (see Appendix B of Ref. [236]) that, whereas on the singlet state the classical correlation function [237] $\frac{2}{\pi }\theta -1$ is linear, the quantum correlations of two are of the “stronger” cosine form −cos(θ). A stronger-than-quantum correlation would be a sign function sgn(θ−π/2) [238].

When translated into the most fundamental empirical level—to two clicks in 2×2=4 respective detectors, a single click on each side—the resulting differences
(19)$$\begin{array}{c} \Delta E=E_{c,2,2}\left(\theta \right)-E_{q,2j+1,2}\left(\theta \right)\\ =-1+\frac{2}{\pi }\theta +\cos\theta =\frac{2}{\pi }\theta +\sum_{k=1}^{\infty }\frac{{\left(-1\right)}^{k}\theta^{2k}}{(2k)!} \end{array}$$
signify a critical difference with regards to the occurrence of joint events: both classical and quantum systems perform the same at the three points $\theta \in \{0,\frac{\pi }{2},\pi \}$. In the region $0<\theta <\frac{\pi }{2}$, ΔE is strictly positive, indicating that quantum mechanical systems “outperform” classical ones with regard to the production of *unequal pairs* “+−” and “−+”, as compared to equal pairs “++” and “−−”. This gets largest at $\theta_{\max}=\arcsin\left(2/\pi \right)\approx 0.69$; at which point the differences amount to 38% of all such pairs, as compared to the classical correlations. Conversely, in the region $\frac{\pi }{2}<\theta <\pi$, ΔE is strictly negative, indicating that quantum mechanical systems “outperform” classical ones with regard to the production of *equal pairs* “++” and “−−”, as compared to unequal pairs “+−” and “−+”. This gets largest at $\theta_{\min}=\pi -\arcsin\left(2/\pi \right)\approx 2.45$. Stronger-than-quantum correlations [239,240] could be of a sign functional form $E_{s,2,2}\left(\theta \right)=\mathrm{sgn}\left(\theta -\pi /2\right)$ [238].

In correlation experiments, these differences are the reason for violations of Boole’s (classical) conditions of possible experience. Therefore, it appears not entirely unreasonable to speculate that the non-classical behavior already is expressed and reflected at the level of these two-particle correlations, and not in need for any violations of the resulting inequalities.

### 6.3. Min-Max Principle

Violation of Boole’s (classical) conditions of possible experience by the quantum probabilities, correlations, and expectations are indications of some sort of non-classicality; and are often interpreted as certification of quantum physics, and quantum physical features [241,242]. Therefore it is important to know the extent of such violations; as well as the experimental configurations (if they exist [243]) for which such violations reach a maximum.

The basis of the min-max method are two observations [244]:Boole’s bounds are *linear*—indeed linearity is, according to Pitowsky [81], the main finding of Boole with regards to *conditions of possible (nowadays classical physical) experience* [45,46]—in the terms entering those bounds, such as probabilities and *n*th order correlations or expectations.All such terms, in particular, probabilities and *n*th order correlations or expectations, have a quantum realization as self-adjoint transformations. As coherent superpositions (linear sums and differences) of self-adjoint transformations are again self-adjoint transformations (and thus normal operators), they are subject to the spectral theorem. Therefore, effectively, all those terms are “bundled together” to give a single “comprehensive” (for Boole’s conditions of possible experience) observable.The spectral theorem, when applied to self-adjoint transformations obtained from substituting the quantum terms for the classical terms, yields an eigensystem consisting of all (pure or non-pure) states, as well as the associated eigenvalues which, according to the quantum mechanical axioms, serve as the measurement outcomes corresponding to the combined, bundled, “comprehensive”, observables. (In the usual Einstein-Podolsky-Rosen “explosion type” setup these quantities will be highly non-local.) The important observation is that this “comprehensive” (for Boole’s conditions of possible experience) observable encodes or includes all possible one-by-one measurements on each one of the single terms alone, at least insofar as they pertain to Boole’s conditions.By taking the minimal and the maximal eigenvalue in the spectral sum of this comprehensive observable one, therefore, obtains the minimal and the maximal measurement outcomes “reachable” by quantization.

Therefore, Boole’s conditions of possible experience are taken as given and for granted, and the computational intractability of their hull problem [80] is of no immediate concern, because nothing needs to be said of actually finding those conditions of possible experience, whose calculation may grow exponentially with the number of vertices. Note also that there might be a possible confusion of the term “min-max principle” [9] (§ 90) with the term “maximal operator’ [9] (§ 84). Finally, this is no attempt to compute general quantum ranges, as for instance discussed by Pitowsky [78,245,246] and Tsirelson [76,247,248].

Indeed, functional analysis provides a technique to compute (maximal) violations of Boole-Bell type inequalities [249,250]: the min-max principle also known as *Courant-Fischer-Weyl min-max principle* for self-adjoint transformations (cf. Ref. [9] (§ 90), Ref. [251] (p. 75ff), and (Section 4.4, p. 142ff in Ref. [252])), or rather an elementary consequence thereof: by the spectral theorem any bounded self-adjoint linear operator T has a spectral decomposition $T=\sum_{i=1}^{n}\lambda_{i}E_{i}$, in terms of the sum of products of bounded eigenvalues times the associated orthogonal projection operators. Suppose for the sake of demonstration that the spectrum is non-degenerate. Then we can (re)order the spectral sum so that $\lambda_{1}\geq \lambda_{2}\geq \ldots \geq \lambda_{n}$ (in case the eigenvalues are also negative, take their absolute value for the sort), and consider the greatest eigenvalue.

In quantum mechanics, the maximal eigenvalue of a self-adjoint linear operator can be identified with the maximal value of an observation. Therefore, the spectral theorem supplies even the state associated with this maximal eigenvalue $\lambda_{1}$: it is the eigenvector (linear subspace) $|e_{1}\rangle$ associated with the orthogonal projector $E_{i}=|e_{1}\rangle \langle e_{1}|$ occurring in the (re)ordered spectral sum of T.

With this in mind, computation of maximal violations of all the Boole-Bell type inequalities associated with Boole’s (classical) conditions of possible experience is straightforward:take all terms containing probabilities, correlations or expectations and the constant real-valued coefficients which are their multiplicative factors; thereby excluding single constant numerical values O(1) (which could be written on “the other” side of the inequality; resulting if what might look like “$T\left(p_{1},\ldots ,p_{n},p_{1,2},\ldots ,p_{123},\ldots \right)\leq O\left(1\right)$” (usually, these inequalities, for reasons of operationalizability, as discussed earlier, do not include highter than 2rd order correlations), and thereby define a function *T*;in the transition “quantization” step T→T substitute all classical probabilities and correlations or expectations with the respective quantum self-adjoint operators, such as for two spin-$\frac{1}{2}$ particles, $p_{1}\to q_{1}=\frac{1}{2}\left[I_{2}\pm \sigma \left(\theta_{1},\varphi_{1}\right)\right]⊗I_{2}$, $p_{2}\to q_{2}=\frac{1}{2}\left[I_{2}\pm \sigma \left(\theta_{2},\varphi_{2}\right)\right]⊗I_{2}$, $p_{12}\to q_{12}=\frac{1}{2}\left[I_{2}\pm \sigma \left(\theta_{1},\varphi_{1}\right)\right]⊗\frac{1}{2}\left[I_{2}\pm \sigma \left(\theta_{2},\varphi_{2}\right)\right]$, $E_{c}\to E_{q}=p_{12++}+p_{12--}-p_{12+-}-p_{12-+}$, as demanded by the inequality. Please note that since the coefficients in T are all real-valued, and because ${\left(A+B\right)}^{†}=A^{†}+B^{†}=\left(A+B\right)$ for arbitrary self-adjoint transformations A,B, the real-valued weighted sum T of self-adjoint transformations is again self-adjoint.Finally, compute the eigensystem of T; in particular the largest eigenvalue $\lambda_{\max}$ and the associated projector which, in the non-degenerate case, is the dyadic product of the “maximal state” $|e_{\max}\rangle$, or $E_{\max}=|e_{\max}\rangle \langle e_{\max}|$.In a last step, maximize $\lambda_{\max}$ (and find the associated eigenvector $|e_{\max}\rangle$) with respect to variations of the parameters incurred in step (ii).

The min-max method yields a feasible, constructive method to explore the quantum bounds on Boole’s (classical) conditions of possible experience. Its application to other situations is feasible. A generalization to higher-dimensional cases appears tedious but with the help of automated formula manipulation straightforward.

#### 6.3.1. Expectations from Quantum Bounds

The quantum expectation can be directly computed from spin state operators. For spin-$\frac{1}{2}$ particles, the relevant operator, normalized to eigenvalues ±1, is
(20)$$\begin{array}{c} T\left(\theta_{1},\varphi_{1};\theta_{2},\varphi_{2}\right)=\left[2S_{\frac{1}{2}}\left(\theta_{1},\varphi_{1}\right)\right]⊗\left[2S_{\frac{1}{2}}\left(\theta_{2},\varphi_{2}\right)\right]. \end{array}$$

The eigenvalues are −1,−1,1,1 and 0; with eigenvectors for $\varphi_{1}=\varphi_{2}=\frac{\pi }{2}$,
(21)$$\begin{array}{c} {\left(-e^{-i(\theta_{1}+\theta_{2})},0,0,1\right)}^{⊺},{\left(0,-e^{-i(\theta_{1}-\theta_{2})},1,0\right)}^{⊺},{\left(e^{-i(\theta_{1}+\theta_{2})},0,0,1\right)}^{⊺},{\left(0,e^{-i(\theta_{1}-\theta_{2})},1,0\right)}^{⊺}, \end{array}$$
respectively.

If the states are restricted to Bell basis states $|\Psi^{\mp }\rangle =\frac{1}{\sqrt{2}}\left(|01\rangle \mp |10\rangle \right)$ and $|\Phi^{\mp }\rangle =\frac{1}{\sqrt{2}}\left(|00\rangle \mp |11\rangle \right)$ and the respective projection operators are $E_{\Psi^{\mp }}$ and $E_{\Phi^{\mp }}$, then the correlations, reduced to the projected operators $E_{\Psi^{\mp }}EE_{\Psi^{\mp }}$ and $E_{\Phi^{\mp }}EE_{\Phi^{\mp }}$ on those states, yield extrema at $-\cos(\theta_{1}-\theta_{2})$ for $E_{\Psi^{-}}$, $\cos(\theta_{1}-\theta_{2})$ for $E_{\Psi^{+}}$, $-\cos(\theta_{1}+\theta_{2})$ for $E_{\Phi^{-}}$, and $\cos(\theta_{1}+\theta_{2})$ for $E_{\Phi^{+}}$.

#### 6.3.2. Quantum Bounds on the House/Pentagon/Pentagram Logic

In a similar way two-particle correlations of a spin-one system can be defined by
(22)$$\begin{array}{c} A\left(\theta_{1},\varphi_{1};\theta_{2},\varphi_{2}\right)=S_{1}\left(\theta_{1},\varphi_{1}\right)⊗S_{1}\left(\theta_{2},\varphi_{2}\right). \end{array}$$

Plugging in these correlations into the Klyachko-Can-Biniciogolu-Shumovsky inequality [88] yields the Klyachko-Can-Biniciogolu-Shumovsky operator
(23)$$\begin{array}{c} \mathrm{KCBS}\left(\theta_{1},\ldots ,\theta_{5},\varphi_{1},\ldots ,\varphi_{5}\right)=A\left(\theta_{1},\varphi_{1},\theta_{3},\varphi_{3}\right)+A\left(\theta_{3},\varphi_{3},\theta_{5},\varphi_{5}\right)\\ +A\left(\theta_{5},\varphi_{5},\theta_{7},\varphi_{7}\right)+A\left(\theta_{7},\varphi_{7},\theta_{9},\varphi_{9}\right)+A\left(\theta_{9},\varphi_{9},\theta_{1},\varphi_{1}\right). \end{array}$$

Taking the special values of Tkadlec [253], which, is spherical coordinates, are $a_{1}\;=\;{\left(1,\frac{\pi }{2},0\right)}^{⊺}$, $a_{2}\;=\;{\left(1,\frac{\pi }{2},\frac{\pi }{2}\right)}^{⊺}$, $a_{3}={\left(1,0,\frac{\pi }{2}\right)}^{⊺}$, $a_{4}={\left(\sqrt{2},\frac{\pi }{2},-\frac{\pi }{4}\right)}^{⊺}$, $a_{5}={\left(\sqrt{2},\frac{\pi }{2},\frac{\pi }{4}\right)}^{⊺}$, $a_{6}\;=\;{\left(\sqrt{6},\tan^{-1}\left(\frac{1}{\sqrt{2}}\right),-\frac{\pi }{4}\right)}^{⊺}$, $a_{7}={\left(\sqrt{3},\tan^{-1}\left(\sqrt{2}\right),\frac{3\pi }{4}\right)}^{⊺}$, $a_{8}={\left(\sqrt{6},\tan^{-1}\left(\sqrt{5}\right),\tan^{-1}\left(\frac{1}{2}\right)\right)}^{⊺}$, $a_{9}={\left(\sqrt{2},\frac{3\pi }{4},\frac{\pi }{2}\right)}^{⊺}$, $a_{10}={\left(\sqrt{2},\frac{\pi }{4},\frac{\pi }{2}\right)}^{⊺}$, yields eigenvalues of KCBS in
(24)$$\begin{array}{c} \left\{-2.49546,2.2288,-1.93988,1.93988,-1.33721,1.33721,-0.285881,0.285881,0.266666\right\} \end{array}$$
all violating the Klyachko-Can-Biniciogolu-Shumovsky inequality [88].

#### 6.3.3. Quantum Bounds on the Cabello, Estebaranz and García-Alcaine Logic

As a final exercise we shall compute the quantum bounds on the Cabello, Estebaranz and García-Alcaine logic [162,189] which can be used in a parity proof of the Kochen-Specker theorem in 4 dimensions, as well as the dichotomic observables (Equation (2) in [100]) $A_{i}=2|a_{i}\rangle \langle a_{i}|-I_{4}$ is used. The observables are then “bundled” into the respective contexts to which they belong; and the context summed according to the non-contextual inequalities from the Hull computation and introduced by Cabello (Equation (1) in [100]). As a result (we use Cabello’s notation and not ours),
(25)$$\begin{array}{c} T=-A_{12}⊗A_{16}⊗A_{17}⊗A_{18}-A_{34}⊗A_{45}⊗A_{47}⊗A_{48}-A_{17}⊗A_{37}⊗A_{47}⊗A_{67}\\ -A_{12}⊗A_{23}⊗A_{28}⊗A_{29}-A_{45}⊗A_{56}⊗A_{58}⊗A_{59}-A_{18}⊗A_{28}⊗A_{48}⊗A_{58}\\ -A_{23}⊗A_{34}⊗A_{37}⊗A_{39}-A_{16}⊗A_{56}⊗A_{67}⊗A_{69}-A_{29}⊗A_{39}⊗A_{59}⊗A_{69} \end{array}$$

The resulting $4^{4}=256$ eigenvalues of T have numerical approximations as ordered numbers −6.94177≤−6.67604≤…≤5.78503≤6.023, neither of which violates the non-contextual inequality enumerated in (Equation (1) in Ref. [100]).

## 7. Epistemologic Deceptions

When reading the book of Nature, she tries to tell us something very sublime yet simple; but what exactly is it? I have the feeling that often discussants approach this particular book not with evenly suspended attention [254,255] but with strong–even ideologic [3] or evangelical [256]–(pre)dispositions. This might be one of the reasons why Specker called this area “haunted” [257]. With these provisos we shall enter the discussion.

Already in 1935—possibly based to the Born rule for computing quantum probabilities which differ from classical probabilities on a global scale involving complementary observables, and yet coincide within contexts—Schrödinger pointed out (cf. also Pitowsky (footnote 2, p. 96 in [81])) that [258] (p. 327) *“at no moment does there exist an ensemble of classical states of the model that squares with the totality of quantum mechanical statements of this moment”. This seems to be the gist of what can be learned from the quantum probabilities: they cannot be accommodated entirely within a classical framework.”*

What can be positively said? There is operational access to a single context (block, maximal observable, orthonormal basis, Boolean subalgebra); and for all that operationally matters, all observables forming that context can be simultaneously value definite. (It could formally be argued that an entire star of contexts intertwined in a “true” proposition must be valued definite.) A single context represents the maximal information encodable into a quantum system. This can be done by state preparation.

Beyond this single context, one can have views on that single state in which the quantized system was prepared. However, these views come at a price: value indefiniteness. (Value indefiniteness is often expressed as contextuality, but this view is distractive, as it suggests some existing entity which is changing its value; depending on how—that is, along which context—it is measured [259].) It might also come as no surprise that by artificially forcing classical relations build on subset relations on Hilbert space entities yields indeterminacies: the standard relation and set theoretical operations on subsets may simply not an adequate framework to investigate vector-worlds.

Moreover, there are grave issues with regards to interpreting whether or not certain detector clicks indicate non-classical performance: the clicks are observed all right; yet what do they mean? (This aspect relates to a discussion [260,261] about whether or not a particular quantum teleportation experiment [262] is achieved only as a postdiction.) Depending on the type of gadget or configuration of observable chosen the same detector click may simultaneously support very different—indeed even mutually contradicting and exclusive—propositions and conclusions [121,148]. This is aggravated by the fact that there are no rules selecting one gadget or cloud of observables over other gadgets or clouds.

This situation might not be taken as a metaphysical conundrum, but perceived rather Socratically: it should come as no surprise that intrinsic [263], emdedded [264] observers have no access to all the information they subjectively desire, but only to a limited amount of properties their system—be it a virtual or a physical universe—is capable of expressing. Therefore there is no omniscience in the wider sense of “all that observers want to know” but rather “all that is operational”.

Indeed, we may have been deceived into believing that all observable we believe to epistemically exist are ontic. Anything beyond this narrow “local omniscience covering a single context” in which the quantized system was prepared appears to be a subjective illusion which is only stochastically supported by the quantum formalism—in terms of Gleason’s “projective views” on that single, value definite context. Experiments may enquire about such value indefinite observables by “forcing” a measurement on a system not prepared or encoded to be interrogated in that way. However, these measurements of non-existing properties, although seemingly possessing viable outcomes which might be interpreted as referring to some alleged hidden properties, cannot carry any (consistent classical) content of that system alone.

To paraphrase a dictum by Peres [237], unprepared contexts do not exist; at least not in any operationally meaningful way. If one nevertheless forces metaphysical existence on (value) indefinite, non-existing, physical entities the price, hedged into the quantum formalism, is stochasticity.

## Figures and Tables

**Figure 1 entropy-22-00602-f001:**
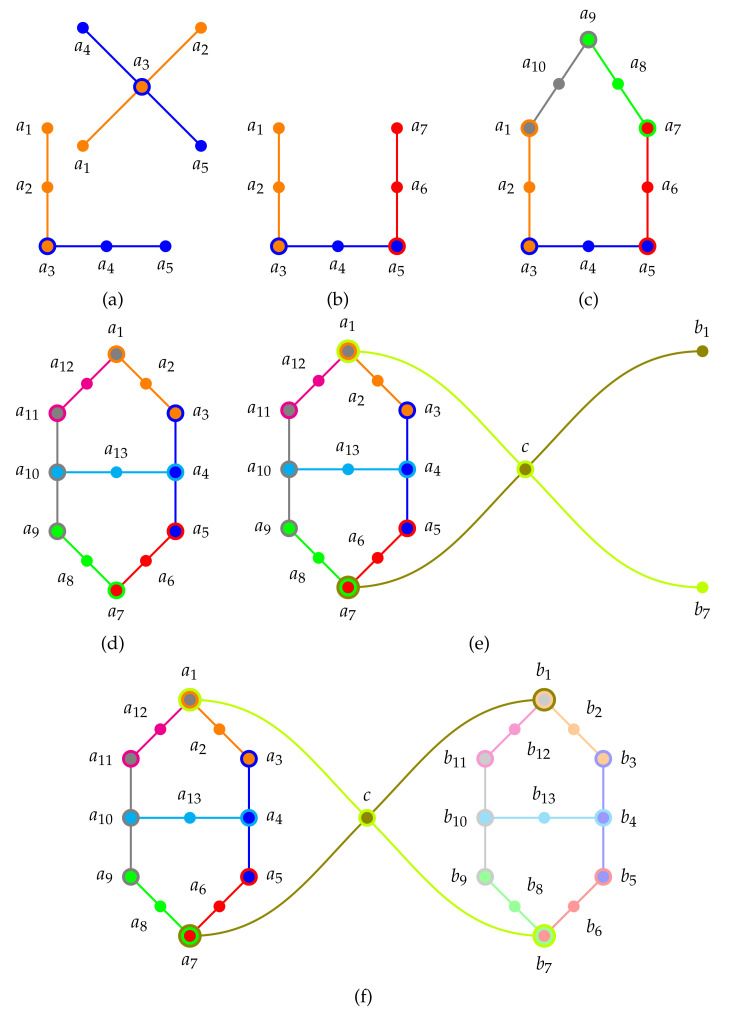
Greechie orthogonality hypergraphs representing historic configurations of complementary observables arranged in 3-element blocks. (**a**) Two renditions of the firefly gadget; (**b**) Two firefly gadgets with a common block; (**c**) house/pentagon/pentagram gadget; (**d**) Specker bug/cat’s cradle logic; (**e**) extended Specker bug logic with a 1-1/true-implies-true property; (**f**) combo of interlinked Specker bugs with a non-separable set of two-valued states (at $a_{1}$-$b_{1}$ as well as $a_{7}$-$b_{7}$).

**Figure 2 entropy-22-00602-f002:**
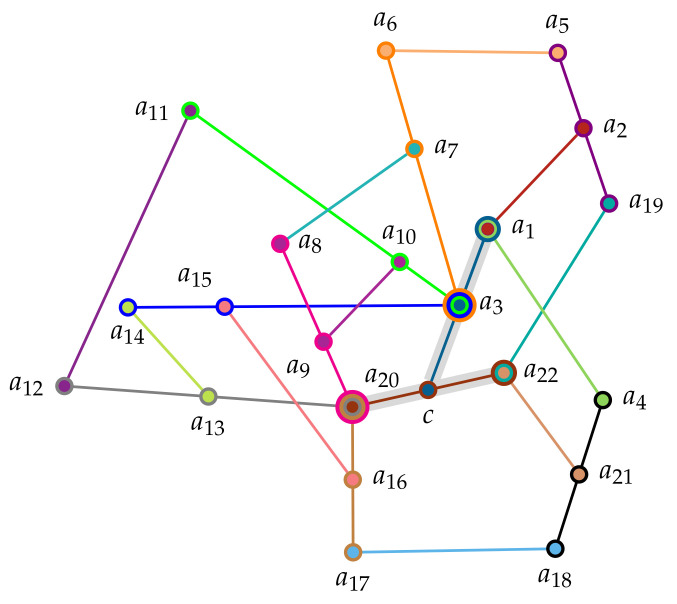
Greechie orthogonality hypergraph [148] (some observables which are not essential to the argument are not drawn) from proof of Theorem 3 in Ref. [119]. The advantage of this true-implies-false gadget is a straightforward parametric faithful orthogonal representation allowing angles $0<∠a_{1},a_{22}\leq \frac{\pi }{4}$ radians ($45^{\circ }$) of, say, the terminal points $a_{1}$ and $a_{22}$. The corresponding logic including the completed set of 34 vertices in 21 blocks is set representable by partition logics because the supported 89 two-valued states are (color) separable. It is not too difficult to prove (by contradiction) that say if both $a_{1}$ as well as $a_{22}$ are assumed to be 1, then $a_{2}$, $a_{3}$, $a_{4}$, as well as $a_{19}$, $a_{20}$ and $a_{21}$ should be 0. Therefore, $a_{5}$ and $a_{18}$ would need to be true. As a result, $a_{6}$ and $a_{17}$ would need to be false. Hence, $a_{7}$ as well as $a_{16}$ would be 1, rendering $a_{8}$ and $a_{15}$ to be 0. This would imply $a_{9}$ as well as $a_{14}$ to be 1, which in turn would demand $a_{10}$ and $a_{13}$ to be false. Therefore, $a_{11}$ and $a_{12}$ would have to be 1, which contradicts the admissibility rules WAD1&WAD2 for value assignments. It is also a true-inplies-true gadget for the terminal points $a_{1}$-$a_{20}$.

**Figure 3 entropy-22-00602-f003:**
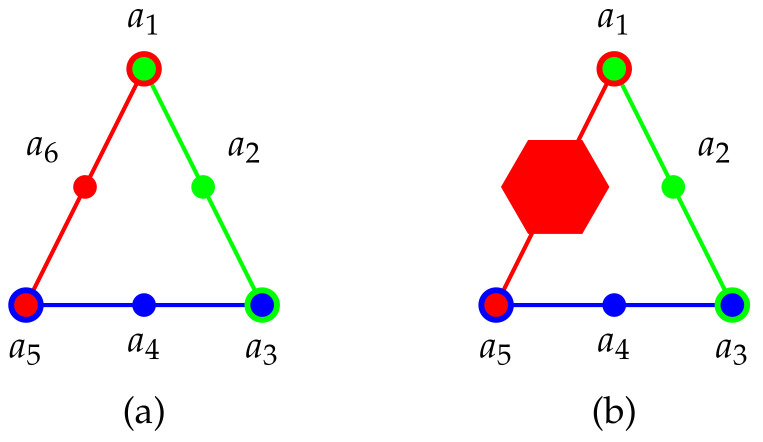
Greechie orthogonality hypergraph (**a**) of the scheme of a true implies true gadget realizable by a partition logic but not as a faithful orthogonal representation in 3-dimensional Hilbert space (in three-dimensional Hilbert space all three “inner” vertices $a_{2}$, $a_{4}$ and $a_{6}$ would “collapse” into any of the three “outer” vertices $a_{1}$, $a_{3}$ or $a_{5}$): suppose $a_{1}=1$; then, according to the admissability axioms WAD1&WAD2, $a_{3}=a_{5}=0$ and thus $a_{4}=1$. (**b**) To achieve a faithful orthogonal representation one could modify this triangular scheme by following Kochen and Specker ([$\Gamma_{1}$, p. 68 in [122]): they substituted one of the three contexts, say $\left\{a_{5},a_{6},a_{1}\right\}$, by a Specker bug which also provides a “true implies false” property at its terminal vertices $a_{1}$ and $a_{5}$, and thereby acts just as admissibility rule WAD1.

**Figure 4 entropy-22-00602-f004:**
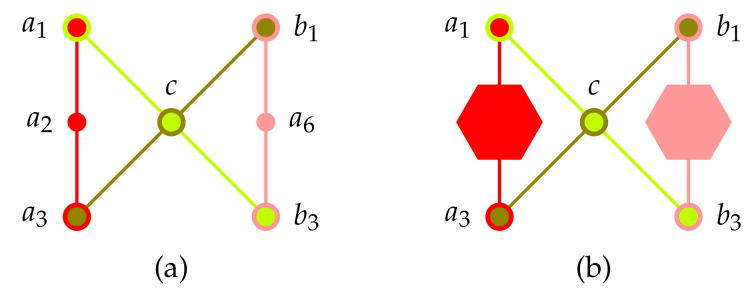
Greechie orthogonality hypergraph (**a**) of the scheme of a logic with a nonseparable set of two-valued states not as a faithful orthogonal representation in 3-dimensional Hilbert space: suppose $a_{1}=1$; then, according to the admissability axioms WAD1&WAD2, $a_{3}=b_{3}=c=0$ and thus $b_{1}=1$ (and, by symmetry, vice versa: $B_{1}=1$ implies $a_{1}=1$). (**b**) To achieve a faithful orthogonal representation one could modify this bowtie scheme by following Kochen and Specker ($\Gamma_{3}$, p. 70 in [122]): they substituted two of the four contexts $\left\{a_{1},a_{2},a_{3}\right\}$ and $\left\{b_{1},b_{2},b_{3}\right\}$ by Specker bugs which also provide a “true implies false” property at their terminal vertices $a_{1}$ and $a_{3}$, as well as $b_{1}$ and $b_{3}$, respectively, and thereby act just as admissibility rule WAD1.

**Figure 5 entropy-22-00602-f005:**
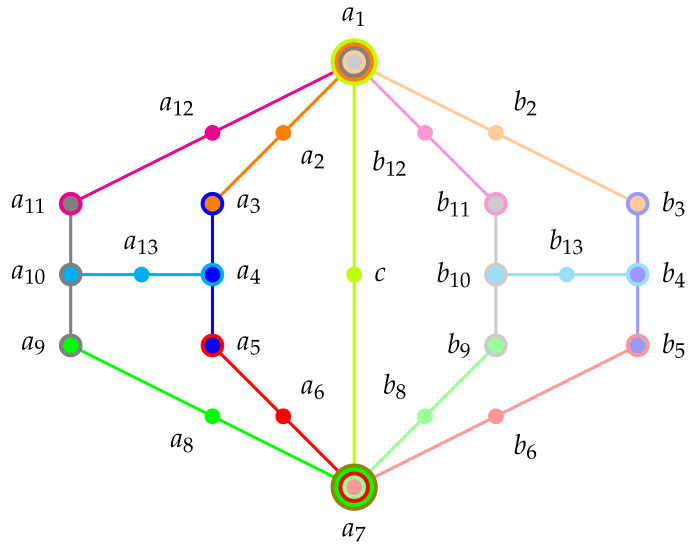
Greechie orthogonality hypergraph of the classical remainder of a combo of interlinked Specker bugs with a non-separable set of two-valued states depicted in Figure 1f.

**Figure 6 entropy-22-00602-f006:**
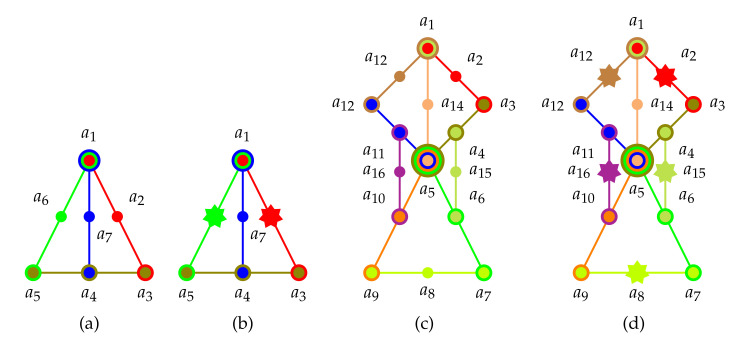
Greechie orthogonality hypergraph schemes of observables with non-unital sets of two-valued states with impossible or unknown faithful orthogonal representations (**a**) simplest scheme without a faithful orthogonal representation: if $v(a_{1})=1$ then $v\left(a_{3}\right)=v\left(a_{4}\right)=v\left(a_{5}\right)=0$ which contradicts admissibility rule WAD1; (**b**) the same as in (**a**) but with $a_{2}$ and $a_{6}$ substituted by a “true-implies-false gadget” (for example, the Specker bug gadget is taken in Figures 2.4 and 6.3 of Ref. [35]); (**c**) another scheme for a logic with a non-unital set of two-valued states: if $v(a_{1})=1$ then $v\left(a_{3}\right)=v\left(a_{5}\right)=v\left(a_{12}\right)=0$; therefore $v\left(a_{4}\right)=v\left(a_{11}\right)=0$; and therefore $v\left(a_{6}\right)=v\left(a_{10}\right)=0$ and $v\left(a_{7}\right)=v\left(a_{9}\right)=1$ which contradicts admissibility rule WAD2; (**d**) the same as in (**a**) but with $a_{2}$ and $a_{6}$ substituted by a “true-implies-false gadget” (for example, the Specker bug gadget is taken in Figure 7.3 of Ref. [35])).

**Table 1 entropy-22-00602-t001:** The 16 two-valued states on the 2 particle two observables per particle configuration.

#	$a_{1}$	$a_{2}$	$a_{3}$	$a_{4}$	$a_{13}$	$a_{14}$	$a_{23}$	$a_{24}$
$v_{1}$	0	0	0	0	0	0	0	0
$v_{2}$	0	0	0	1	0	0	0	0
$v_{3}$	0	0	1	0	0	0	0	0
$v_{4}$	0	0	1	1	0	0	0	0
$v_{5}$	0	1	0	0	0	0	0	0
$v_{6}$	0	1	0	1	0	0	0	1
$v_{7}$	0	1	1	0	0	0	1	0
$v_{8}$	0	1	1	1	0	0	1	1
$v_{9}$	1	0	0	0	0	0	0	0
$v_{10}$	1	0	0	1	0	1	0	0
$v_{11}$	1	0	1	0	1	0	0	0
$v_{12}$	1	0	1	1	1	1	0	0
$v_{13}$	1	1	0	0	0	0	0	0
$v_{14}$	1	1	0	1	0	1	0	1
$v_{15}$	1	1	1	0	1	0	1	0
$v_{16}$	1	1	1	1	1	1	1	1

**Table 2 entropy-22-00602-t002:** Two-valued states on the firefly logic.

#	$a_{1}$	$a_{2}$	$a_{3}$	$a_{4}$	$a_{5}$
$v_{1}$	0	0	0	0	1
$v_{2}$	0	1	0	1	0
$v_{3}$	0	1	1	0	0
$v_{4}$	1	0	1	0	0
$v_{5}$	1	0	0	1	0

**Table 3 entropy-22-00602-t003:** (Color online) Two-valued states on the pentagon.

#	$a_{1}$	$a_{2}$	$a_{3}$	$a_{4}$	$a_{5}$	$a_{6}$	$a_{7}$	$a_{8}$	$a_{9}$	$a_{10}$
$v_{1}$	1	0	0	1	0	1	0	1	0	0
$v_{2}$	1	0	0	0	1	0	0	1	0	0
$v_{3}$	1	0	0	1	0	0	1	0	0	0
$v_{4}$	0	0	1	0	0	1	0	1	0	1
$v_{5}$	0	0	1	0	0	0	1	0	0	1
$v_{6}$	0	0	1	0	0	1	0	0	1	0
$v_{7}$	0	1	0	0	1	0	0	1	0	1
$v_{8}$	0	1	0	0	1	0	0	0	1	0
$v_{9}$	0	1	0	1	0	0	1	0	0	1
$v_{10}$	0	1	0	1	0	1	0	0	1	0
$v_{11}$	0	1	0	1	0	1	0	1	0	1
$v_{e}$	$\frac{1}{2}$	0	$\frac{1}{2}$	0	$\frac{1}{2}$	0	$\frac{1}{2}$	0	$\frac{1}{2}$	0

**Table 4 entropy-22-00602-t004:** The 14 two-valued states on the Specker bug (cat’s cradle) logic.

#	$a_{1}$	$a_{2}$	$a_{3}$	$a_{4}$	$a_{5}$	$a_{6}$	$a_{7}$	$a_{8}$	$a_{9}$	$a_{10}$	$a_{11}$	$a_{12}$	$a_{13}$
$v_{1}$	1	0	0	0	1	0	0	0	1	0	0	0	1
$v_{2}$	1	0	0	1	0	1	0	0	1	0	0	0	0
$v_{3}$	1	0	0	0	1	0	0	1	0	1	0	0	0
$v_{4}$	0	1	0	0	1	0	0	0	1	0	0	1	1
$v_{5}$	0	1	0	0	1	0	0	1	0	0	1	0	1
$v_{6}$	0	1	0	1	0	1	0	0	1	0	0	1	0
$v_{7}$	0	1	0	1	0	0	1	0	0	0	1	0	0
$v_{8}$	0	1	0	1	0	1	0	1	0	0	1	0	0
$v_{9}$	0	1	0	0	1	0	0	1	0	1	0	1	0
$v_{10}$	0	0	1	0	0	0	1	0	0	0	1	0	1
$v_{11}$	0	0	1	0	0	1	0	1	0	0	1	0	1
$v_{12}$	0	0	1	0	0	1	0	0	1	0	0	1	1
$v_{13}$	0	0	1	0	0	0	1	0	0	1	0	1	0
$v_{14}$	0	0	1	0	0	1	0	1	0	1	0	1	0

**Table 5 entropy-22-00602-t005:** The 24 two-valued states on the interconnected Specker combo $\Gamma_{3}$ [122] (p. 70).

#	$a_{1}$	$a_{2}$	$a_{3}$	$a_{4}$	$a_{5}$	$a_{6}$	$a_{7}$	$a_{8}$	$a_{9}$	$a_{10}$	$a_{11}$	$a_{12}$	$a_{13}$	$b_{1}$	$b_{2}$	$b_{3}$	$b_{4}$	$b_{5}$	$b_{6}$	$b_{7}$	$b_{8}$	$b_{9}$	$b_{10}$	$b_{11}$	$b_{12}$	$b_{13}$	*c*
$v_{1}$	1	0	0	1	0	1	0	0	1	0	0	0	0	1	0	0	0	1	0	0	0	1	0	0	0	1	0
$v_{2}$	1	0	0	0	1	0	0	1	0	1	0	0	0	1	0	0	0	1	0	0	0	1	0	0	0	1	0
$v_{3}$	1	0	0	0	1	0	0	0	1	0	0	0	1	1	0	0	1	0	1	0	0	1	0	0	0	0	0
$v_{4}$	1	0	0	0	1	0	0	0	1	0	0	0	1	1	0	0	0	1	0	0	1	0	1	0	0	0	0
$v_{5}$	0	1	0	1	0	1	0	1	0	0	1	0	0	0	1	0	1	0	1	0	1	0	0	1	0	0	1
$v_{6}$	0	1	0	1	0	1	0	1	0	0	1	0	0	0	1	0	1	0	1	0	0	1	0	0	1	0	1
$v_{7}$	0	1	0	1	0	1	0	1	0	0	1	0	0	0	1	0	0	1	0	0	1	0	1	0	1	0	1
$v_{8}$	0	1	0	1	0	1	0	1	0	0	1	0	0	0	0	1	0	0	1	0	1	0	1	0	1	0	1
$v_{9}$	0	1	0	1	0	1	0	0	1	0	0	1	0	0	1	0	1	0	1	0	1	0	0	1	0	0	1
$v_{10}$	0	1	0	1	0	1	0	0	1	0	0	1	0	0	1	0	1	0	1	0	0	1	0	0	1	0	1
$v_{11}$	0	1	0	1	0	1	0	0	1	0	0	1	0	0	1	0	0	1	0	0	1	0	1	0	1	0	1
$v_{12}$	0	1	0	1	0	1	0	0	1	0	0	1	0	0	0	1	0	0	1	0	1	0	1	0	1	0	1
$v_{13}$	0	1	0	1	0	0	1	0	0	0	1	0	0	0	0	1	0	0	0	1	0	0	0	1	0	1	0
$v_{14}$	0	1	0	0	1	0	0	1	0	1	0	1	0	0	1	0	1	0	1	0	1	0	0	1	0	0	1
$v_{15}$	0	1	0	0	1	0	0	1	0	1	0	1	0	0	1	0	1	0	1	0	0	1	0	0	1	0	1
$v_{16}$	0	1	0	0	1	0	0	1	0	1	0	1	0	0	1	0	0	1	0	0	1	0	1	0	1	0	1
$v_{17}$	0	1	0	0	1	0	0	1	0	1	0	1	0	0	0	1	0	0	1	0	1	0	1	0	1	0	1
$v_{18}$	0	0	1	0	0	1	0	1	0	1	0	1	0	0	1	0	1	0	1	0	1	0	0	1	0	0	1
$v_{19}$	0	0	1	0	0	1	0	1	0	1	0	1	0	0	1	0	1	0	1	0	0	1	0	0	1	0	1
$v_{20}$	0	0	1	0	0	1	0	1	0	1	0	1	0	0	1	0	0	1	0	0	1	0	1	0	1	0	1
$v_{21}$	0	0	1	0	0	1	0	1	0	1	0	1	0	0	0	1	0	0	1	0	1	0	1	0	1	0	1
$v_{22}$	0	0	1	0	0	0	1	0	0	1	0	1	0	0	0	1	0	0	0	1	0	0	0	1	0	1	0
$v_{23}$	0	0	1	0	0	0	1	0	0	0	1	0	1	0	1	0	1	0	0	1	0	0	0	1	0	0	0
$v_{24}$	0	0	1	0	0	0	1	0	0	0	1	0	1	0	0	1	0	0	0	1	0	0	1	0	1	0	0

**Table 6 entropy-22-00602-t006:** The 6 two-valued states on the non-unital Tkadlec logic (Figure 2, p. 20 in [143]).

#	$a_{1}$	$a_{2}$	$a_{3}$	… … … … … … … … …																																$a_{36}$
$v_{1}$	0	1	0	1	0	0	0	1	0	0	0	0	1	0	0	0	0	0	1	0	0	0	0	0	1	1	1	1	1	1	0	0	1	1	1	1
$v_{2}$	0	1	0	1	0	0	0	0	0	0	1	0	0	0	0	1	0	1	0	0	0	0	1	0	0	1	1	0	1	1	1	1	1	0	1	0
$v_{3}$	0	1	0	1	0	0	0	0	0	0	0	1	0	0	0	1	0	1	0	0	0	0	1	0	1	1	1	0	0	1	1	0	1	1	1	0
$v_{4}$	0	1	0	0	1	1	0	0	0	0	0	1	0	1	0	1	0	0	0	0	0	0	1	0	0	1	1	1	0	0	1	1	1	1	0	0
$v_{5}$	0	1	0	0	1	1	0	0	0	0	0	1	0	0	1	0	0	0	0	0	0	0	1	0	0	0	1	1	0	0	1	1	1	1	1	1
$v_{6}$	0	1	0	0	1	0	1	0	0	0	0	0	1	1	0	0	0	0	0	0	1	0	0	0	0	1	0	1	1	1	1	1	0	1	0	1

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
