# Peer review of "What Is So Special about Quantum Clicks?"

_entropy, 2020, doi:10.3390/e22060602_

Round 1
Reviewer 1 Report
The paper presents an extensive and detailed discussion of the way in which the quantum description of the world departures from the classical one. Although, being physicist, I feel that syntax slightly prevails here semantics I consider the paper as valuable contribution containing a number of interesting observations from the borderline of physics, mathematics, logics and philosophy. Its value is increased by the rich bibliography which I find very useful. I have learned some interesting things like, for example, on the progress concerning the simplification of the original proof of Kochen-Specker theorem or the relation between the chromatic number of the unit sphere, Gleason's theorem and Kochen-Specker theorem. I also appreciate author's sense of humor:"...at that time the Swiss may have inhabitated their biotype of quantum logical thinking..".
I recommend publication.
Author Response
I kindly thank the Referee for the attention and efforts, and also for valuing the sentence cited.
Reviewer 2 Report
I consider the paper as suitable for publication in Entropy. It contains an extensive discussion of the peculiar properties of the quantum description of physical world as contrasted with the classical description. Author discusses some important mathematical constructions stressing their linear character and comparing them (and contrasting with) to those related to Boolear algebra underlying the logic of classical physics. The paper proposes some bird’s-eye view on the mathematical, physical and logical problems inherent in quantum mechanical description. An important part of the paper is an extensive bibliography covering the majority of most important contributions to the subject (including even some papers by Freud!).
Author Response
I kindly thank the Referee for the attention and efforts, also for the esteem with regard to the reference of a quote by Freud.
Reviewer 3 Report
In the reviewed manuscript, the author considers a variety of issues, both in terms of single outcomes as well as probabilistic predictions, supporting the “extra” advantage of the performance of quantized physical systems over classical ones. He, in particular, gives the detailed analysis of general principles for object/observable construction; general principles for probabilities of objects/observables; general framework for computing Boole’s conditions of possible experience; analysis of quantum predictions: probabilities and expectations, etc. The author detailed presentation makes a valuable contribution to understanding of many quantum versus classical problems.
I recommend the manuscript for publication in its present form
Author Response
I kindly thank the Referee for the attention and efforts.